# SINGLETON-OPTIMIZED CONFORMAL PREDICTION

**Tao Wang**[1] *  **Yan Sun**[2]   **Edgar Dobriban**[1]
[1] University of Pennsylvania   [2] New Jersey Institute of Technology
{tawan, dobriban}@wharton.upenn.edu  {yan.sun}@njit.edu

## ABSTRACT

Conformal prediction can be used to construct prediction sets that cover the true outcome with a desired probability, but can sometimes lead to large prediction sets that are costly in practice. The most useful outcome is a singleton prediction—an unambiguous decision—yet existing efficiency-oriented methods primarily optimize average set size. Motivated by this, we propose a new nonconformity score that aims to minimize the probability of producing non-singleton sets. Starting from a non-convex constrained optimization problem as a motivation, we provide a geometric reformulation and associated algorithm for computing the nonconformity score and associated split conformal prediction sets in $O(K)$ time for $K$-class problems. Using this score in split conformal prediction leads to our proposed Singleton-Optimized Conformal Prediction (SOCOP) method. We evaluate our method in experiments on image classification and LLM multiple-choice question-answering, comparing with standard nonconformity scores such as the (negative) label probability estimates and their cumulative distribution function; both of which are motivated by optimizing length. The results show that SOCOP increases singleton frequency (sometimes by over 20%) compared to the above scores, with minimal impact on average set size.

## 1 INTRODUCTION

Reliable uncertainty quantification is often needed for deploying predictive models in settings of importance. While standard single point predictions can be very useful if models are accurate, they can be problematic if model accuracy drops. Prediction sets address this limitation by providing a subset of possible labels, $C(x) \subseteq \mathcal{Y}$, for a given input $x \in \mathcal{X}$. The primary requirement for such sets is usually a form of *coverage*. Formally, given features $X \in \mathcal{X}$ with some distribution, and a multi-class label $Y \in \mathcal{Y}$, we seek sets $C(X) \subseteq \mathcal{Y}$ satisfying the marginal coverage guarantee $\mathbb{P}\{Y \in C(X)\} \geq 1 - \alpha$. Conformal prediction (see e.g., Vovk et al., 1999; Gammerman et al., 1998; Vovk et al., 2005, etc) offers a methodology for constructing prediction sets that satisfy this guarantee under the mild assumption of data exchangeability.

While validity is essential, the practical utility of a prediction set is determined by its *efficiency*. For instance, a trivial set containing all labels is valid but uninformative. In practice, efficiency is often evaluated by the expected size of the sets $\mathbb{E}_X[|C(X)|]$. A variety of works have studied how to achieve small sets on average, ranging from choosing suitable nonconformity scores to explicit optimization approaches (see e.g., Takeuchi, 2020; Sadinle et al., 2019; Romano et al., 2020; Angelopoulos et al., 2021; Kiyani et al., 2024, etc).

However, average size is not necessarily the ideal measure of efficiency. Often, the most desirable outcome is an unambiguous prediction, a *singleton set* [1] containing only one label. A set of size two or more may require additional human intervention or changing the workflow when used in downstream analysis, and thus brings an outsized cost. This motivates an alternative efficiency criterion, first conceptualized in Vovk et al. (2005) as the M-criterion, which seeks to minimize the probability

---

*Correspondence to: tawan@wharton.upenn.edu, dobriban@wharton.upenn.edu

[1]Strictly speaking, a singleton set refers to a cardinality of exactly one ($|C(X)| = 1$). In this work, we use the term "singleton objective" to broadly refer to the goal of minimizing the probability of returning multiple labels ($|C(X)| > 1$). As we discuss below in our experiments, zero sets occur extremely rarely, and so the two objectives effectively coincide.

of producing a non-singleton set, $\mathbb{P}_X[|C(X)| > 1]$. We refer to this as the *singleton objective*. To our knowledge, practical conformal prediction methods that aim to optimize the singleton objective have not yet been developed.

In this work, we bridge this gap by developing conformal prediction sets motivated by optimizing a combination of the singleton objective and the expected length for classification problems, subject to coverage. We begin by formulating this as an optimization problem over prediction sets (which are discrete variables). Our main contributions are then as follows:

1. **Nonconformity score inspired by singleton objective.** We use the singleton objective as inspiration to define a nonconformity score aiming to enhance singleton probability. Since the original optimization problem is constrained, we consider its Lagrangian, which we show is separable across $x$. We show that for each fixed $x$, the optimal prediction set is the set of top-few labels, and that the prediction sets are nested as the Lagrangian penalty parameter increases. This motivates us to define a nonconformity score based on nested conformal prediction (Vovk et al., 2005; Gupta et al., 2022).

2. **Efficient algorithm to compute nonconformity score:** We derive a highly efficient algorithm to compute the nonconformity score, through a geometric perspective. We show that this problem reduces to finding the lower convex hull of a set of $K$ two-dimensional points for $K$-class classification problems, which has $O(K)$ complexity per instance. We show that split conformal prediction sets can be computed with the same complexity.

3. **Empirical validation:** We conduct detailed experiments on three image classification datasets (two versions of ImageNet and TissueMNIST) and LLM multiple-choice question answering. The results demonstrate that our method, which we call Singleton-Optimized Conformal Prediction (SOCOP), achieves a favorable balance between minimizing average set size and maximizing the frequency of singleton predictions compared to state-of-the-art baselines. Often, we can reduce the non-singleton probability by a large fraction (such as 20%) while only incurring a small increase in expected set size.

**Notation.** For a positive integer $K$, we denote $[K] := \{1, \ldots, K\}$. We denote the $(K-1)$-dimensional simplex of probabilities by $\Delta_{K-1} := \{(z_1, \ldots, z_K) : \sum_{i=1}^{K} z_i = 1\}$. For a finite set $A$, we write $|A|$ for its cardinality. The indicator of a set $A$ is denoted by $I(A)$.

## 1.1 RELATED WORK

The origins of distribution-free prediction sets date back to the early works of Wilks (1941), Wald (1943), Scheffe & Tukey (1945), and Tukey (1947; 1948). Distribution-free inference and conformal prediction has been extensively studied in recent works (see, e.g., Saunders et al., 1999; Vovk et al., 1999; Papadopoulos et al., 2002; Vovk et al., 2005; Vovk, 2013; Lei et al., 2013; Lei & Wasserman, 2014; Lei et al., 2018; Romano et al., 2020, etc). Overviews of the field are provided by Vovk et al. (2005); Shafer & Vovk (2008), and Angelopoulos & Bates (2023).

Recent research has started investigating ways to improve the efficiency of prediction sets. (Sadinle et al., 2019) have shown that the true probability of the labels given the features is the conformity score that leads to prediction sets that minimize expected length. Adaptive scoring schemes (Romano et al., 2020; Angelopoulos et al., 2021) have a similar motivation, but are derived from a conditional coverage perspective. These works are related to ours in that we also derive a new nonconformity score. However, taking into account the singleton probability or M-criterion (Vovk et al., 2005), our work requires addressing new technical challenges in terms of efficiently computing the prediction sets. Recent work aims to directly optimize the length, possibly with conditional coverage guarantees (Kiyani et al., 2024). Other work has explored different notions of efficiency, through direct optimization (Stutz et al., 2022; Shi et al., 2025), computational shortcuts (Liang et al., 2023), or other approaches, see e.g., Liang et al. (2025); Le Bars & Humbert (2025); Braun et al. (2025); Behboodi et al. (2025), etc. Additional related work is discussed in Appendix A.

## 2 A SINGLETON-OPTIMIZED NONCONFORMITY SCORE

We begin with a population-level optimization formulation that motivates our nonconformity score.

## 2.1 Problem Formulation

We consider a classification problem with labels $y \in \mathcal{Y} = \{1, \ldots, K\}$ and features $x \in \mathcal{X} = \mathbb{R}^d$. Our goal is to construct prediction sets $C(x)$, for all $x$, satisfying the coverage guarantee $P(Y \in C(X)) \geqslant 1 - \alpha$. Let $\mathcal{M}$ be the collection of all[2] (measurable) prediction sets $C : \mathcal{X} \to 2^{\mathcal{Y}}$. Our motivating problem is to find prediction sets that are optimal with respect to a linear combination of the singleton objective and length, subject to coverage:

$$\min_{C \in \mathcal{M}} \quad F_\lambda(C) := \mathbb{P}_X \left[ |C(X)| > 1 \right] + \lambda \mathbb{E}_X \left[ |C(X)| \right]$$

$$\text{s.t.} \quad G(C) := \mathbb{P}(Y \in C(X)) - (1 - \alpha) \geqslant 0.$$

where $\lambda \geqslant 0$ is a regularization parameter that we will set later. This objective balances the probability of non-singletons $\mathbb{P}_X \left[ |C(X)| > 1 \right]$ and the expected size $\mathbb{E}_X \left[ |C(X)| \right]$. We will argue that this leads to a favorable trade-off, whereby increasing one by a small amount results in a large decrease in the other.

This optimization problem is defined over prediction sets, which belong to a discrete, discontinuous space (e.g., the linear combination of two sets is undefined), and so standard gradient-based optimization methods are not applicable. However, we emphasize that this problem will merely serve as a motivation for us to define a useful nonconformity score. We will not attempt to solve this problem exactly, but rather use it as a starting point, transforming it into a form that allows us to derive our nonconformity score.

Our first step towards defining the nonconformity score is to study the dual of the above problem. This will allow us to use separability in the solution, and thus derive a nonconformity score. Let $P_X$ be the distribution of $X$. The Lagrangian with dual variable $\eta \geqslant 0$ is:

$$\mathcal{L}_\lambda(C, \eta) = \int_{\mathcal{X}} \left[ I(|C(x)| > 1) + \lambda |C(x)| - \eta \sum_{y \in C(x)} P_{Y|X}(y|x) \right] P_X(\mathrm{d}x) + \eta(1 - \alpha). \quad (1)$$

Since $\mathcal{L}_\lambda(C, \eta) = F_\lambda(C) - \eta G(C) \leqslant F_\lambda(C)$ for every feasible $C$ and $\eta \geqslant 0$, minimizing $\mathcal{L}_\lambda(C, \eta)$ gives a lower bound on the original problem.[3]

A key observation is that the minimization of $\mathcal{L}_\lambda(C, \eta)$ over $C$ is separable in $x$, i.e., it can be solved by optimizing over each $x$ separately. Denote, for all $x \in \mathcal{X}$, the per-instance loss

$$\ell_{p(\cdot|x), \lambda} \left( C(x); \eta \right) = I(|C(x)| > 1) + \lambda |C(x)| - \eta \sum_{y \in C(x)} p(y|x).$$

Then, we can write $\mathcal{L}_\lambda(C, \eta) = \int_{\mathcal{X}} \ell_{p(\cdot|x), \lambda} \left( C(x); \eta \right) P_X(\mathrm{d}x) + \eta(1 - \alpha)$ as an integral of the per-instance loss. Thus, (1) can be minimized over $C \in \mathcal{M}$ by minimizing $\ell_{p(\cdot|x), \lambda} \left( C(x); \eta \right)$ for each $x \in \mathcal{X}$ separately. Since $\ell_{p(\cdot|x), \lambda} \left( C(x); \eta \right)$ can be viewed as an instance-level cost associated with the prediction set $C(x)$ and the probabilities of the labels $p(\cdot|x)$, this motivates us to leverage it to construct our nonconformity score.

Continuing with the general approach of leveraging the theoretically optimal prediction set for the construction of the nonconformity score, we study the minimization of $\ell$. For any probability distribution $\gamma \in \Delta_{K-1}$, and Lagrange multiplier $\eta \geqslant 0$, we consider solving for the following *singleton-optimized set* $S_{\eta, \gamma} \subseteq [K]$, defined by the optimization problem[4]

$$S_{\eta, \gamma} := S_{\eta, \gamma, \lambda} \in \arg \min_{S \subseteq \mathcal{Y}} \ell_{\gamma, \lambda}(S; \eta). \quad (2)$$

---

[2]We will endow $\mathcal{X}$ with the Borel $\sigma$-algebra. All quantities considered in this paper will be measurable with respect to appropriate $\sigma$-algebras; this will not be mentioned further.

[3]An optimal solution $C^*$ to this problem minimizes the original objective $F$ subject to the constraint $\mathbb{P}(Y \in C(X)) = \mathbb{P}(Y \in C^*(X))$. For this reason, it would be reasonable to consider the original optimization problem subject to the constraint $G(C) = 0$, in which case, the Lagrange multiplier approach could provide a certificate of optimality or near-optimality quite directly. However, ultimately, we will not solve the above problem directly but rather only use it as a way to define a nonconformity score, which we will then use in conformal prediction. Therefore, certifying the optimality of our intermediate solution to the original optimization problem is not a central goal of our research.

[4]If there are multiple solutions, we choose any set that has a minimal size. The same holds for the definitions in the following text. Our claims will hold for all optimizing sets, and for simplicity we will refer to "the" optimizer.

Then, all solutions of minimizing (1) can be written[5] as $C_\eta(x) := S_{\eta, p(\cdot|x)}$.

## 2.2 DEFINITION OF THE NONCONFORMITY SCORE

From now on, without loss of generality, we order the probabilities such that $\gamma_{y_1} \geq \gamma_{y_2} \geq \cdots \geq \gamma_{y_K} > 0$, where $K = |\mathcal{Y}|$. Fortunately, the structure of the prediction sets $S_{\eta, \gamma}$ can be characterized. A starting point is the following simple result, whose proof (with all proofs) is provided in the appendix. For any $j \in \{0, 1, \ldots, K\}$, let $\mathcal{F}_j$ denote a set of the top $j$ labels, breaking ties arbitrarily; where $\mathcal{F}_0$ is the empty set.

**Lemma 2.1** (The structure of singleton optimal sets). *For any $\eta \geqslant 0$ and $\gamma \in \Delta_{K-1}$, $S_{\eta, \gamma}$ is the set of top-$j$ labels for some $j$ that depends on $\eta$ and $\gamma$.*

The next and crucial observation is that the sets $S_{\eta, \gamma}$ from (2) are *nested* as a function of the Lagrange multiplier $\eta$.

**Lemma 2.2** (Nested Sets Property). *For $0 \leqslant \eta_1 < \eta_2$, we have $S_{\eta_1, \gamma} \subseteq S_{\eta_2, \gamma}$.*

This motivates us to define a nonconformity score via nested conformal prediction (Vovk et al., 2005; Gupta et al., 2022), where we aim to find the smallest $\eta$—and thus the smallest set $S_{\eta, \gamma}$—that contains the true label.

In practice, the true conditional probability $p(\cdot|x)$ is typically unknown; and instead, we only have access to an estimated probability $\hat{p}(\cdot|x)$. By plugging in the estimated probabilities in lieu of the true ones and using nested conformal prediction (Vovk et al., 2005; Gupta et al., 2022), we define the singleton-optimized nonconformity score:

**Definition 2.3** (Singleton-optimized nonconformity score). *For an input $x \in \mathcal{X}$ with label $y \in \mathcal{Y}$, for a probabilistic predictor $\hat{p}$ such that $\hat{p}(\cdot \mid x)$ is a probability distribution over $\mathcal{Y}$, and a regularization parameter $\lambda \geqslant 0$, define the* singleton-optimized nonconformity score

$$r(x, y) := r_\lambda(x, y) = \inf \left\{ \tau \geq 0 : y \in S_{\tau, \hat{p}(\cdot|x), \lambda} \right\}. \tag{3}$$

*where the singleton-optimal set $S_{\eta, \gamma, \lambda}$ is defined in (2) for a Lagrange multiplier $\eta \geqslant 0$.*

In principle, this nonconformity score can be used with a variety of techniques from conformal prediction, including split conformal prediction (Papadopoulos et al., 2002), cross-conformal prediction (Vovk, 2015), Mondrian and label conditional conformal prediction (Vovk et al., 2005), etc, to construct prediction sets. The method of choice depends on the type of data and guarantee desired. However, the practical use of the nonconformity score first requires an efficient algorithm to compute it. As we will see below, a naive search over $\tau$ can be expensive when the number of classes is large. In what follows, we discuss how to compute the nonconformity score $r$ efficiently. Readers more interested in experimental results may skip to Section 3.

## 2.3 GEOMETRIC APPROACH TO COMPUTING THE NONCONFORMITY SCORE

In order to develop a method to compute the nonconformity score, we first study the problem of computing the prediction set $S_{\eta, \gamma}$ for a given vector of probabilities $\gamma \in \Delta_{K-1}$. This is used directly in the nonconformity score. By Lemma 2.1, the optimal prediction sets from (2) are equal to the top few labels. Specifically, $S_{\eta, \gamma} = \mathcal{F}_{\kappa(\eta; \gamma)}$, where $\kappa(\eta; \gamma)$ is the *optimal subset size* (or *optimal index*), defined via the optimization problem:

$$\kappa(\eta; \gamma) := \arg \min_{0 \leq k \leq K} \left\{ \Psi_\eta(k, \gamma) := I(k > 1) + \lambda k - \eta \cdot \sum_{i=1}^{k} \gamma_{y_i} \right\}. \tag{4}$$

For a fixed value of $\eta$, the optimal index $\kappa(\eta; \gamma)$ can be found in time $O(K)$ by observing that, for $k \geqslant 3$, the gaps $\delta_k := \Psi_\eta(k, \gamma) - \Psi_\eta(k-1, \gamma) = \lambda - \eta\gamma_{y_k}$ are non-decreasing in $k$ due to the ordering $\gamma_{y_1} \geqslant \gamma_{y_2} \geqslant \ldots$. Hence, to find the optimum, it is enough to find the smallest index $k^* \geqslant 3$ such that $\delta_{k^*} < 0 \leqslant \delta_{k^*+1}$, if such an index exists; otherwise setting $k^* = K$. Then, we compare the objective value at $k^*$ with those for $k = 0, 1, 2$ and choose the best. This immediately leads to an $O(K)$ algorithm for computing the prediction set $S_{\eta, \gamma}$.

---

[5]When the value of $\lambda$ is fixed or clear from the context, we will often omit it from our notation.

Next, by leveraging the reduced problem (4), the nonconformity score in (3) can be equivalently written as:

$$r(x, y_i) = \inf\{\tau \geqslant 0 : \kappa(\tau; \hat{p}(\cdot|x)) \geqslant i\}. \tag{5}$$

A direct approach might be to search over values of $\tau$, checking $\kappa(\tau; \hat{p}(\cdot|x)) \geqslant i$ for each case, until we find a value that approximates the true value within a certain desired accuracy. However, this direct approach becomes computationally challenging for large values of $K$, because computing the optimal index takes linear time $O(K)$ for each $\tau$. Therefore we propose a fast alternative computational method, which relies on studying the optimal index for different values of $\tau$ simultaneously, and can be viewed through a geometric perspective.

A first step observation is that the nested sets property immediately implies that $\eta \mapsto \kappa(\eta; \gamma)$ is a monotone step function.

**Corollary 2.4** (Properties of optimal index function). *For any $\gamma \in \Delta_{K-1}$, $\kappa(\cdot; \gamma) : [0, \infty] \to \{0, 1, ..., K\}$ is a monotonically non-decreasing, left-continuous step function with $\kappa(0; \gamma) = 0$ and $\kappa(\infty; \gamma) := \lim_{\eta \to \infty} \kappa(\cdot; \gamma) = K$.*

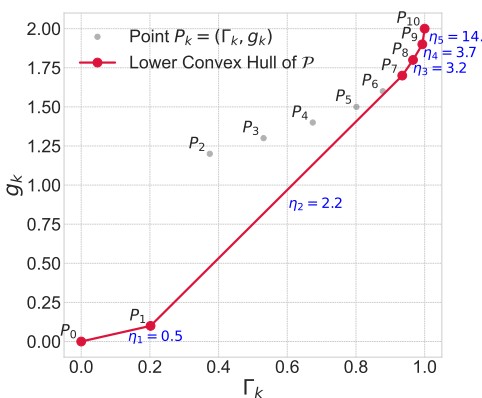

Figure 1: Lower convex hull for a simulated probability vector with $K = 10$.

Next, we aim to characterize the specific points where the jumps of $\kappa$ happen. Denote $\Gamma_k = \sum_{i=1}^{k} \gamma_{y_i}$ and $g_k = I(k > 1) + \lambda k$ for conciseness. For each $k = 0, 1, \ldots$, we consider the point $P_k = (\Gamma_k, g_k)$ in $\mathbb{R}^2$. This yields a set of $K + 1$ points $\mathcal{P} = \{P_0, \ldots, P_K\}$. Our algorithm will leverage the convex hull of $\mathcal{P}$, i.e., $\{\sum_{i=0}^{K} \beta_i P_i : \beta_i \geqslant 0, \sum_{i=0}^{K} \beta_i = 1\}$, which is a convex polygon in $\mathbb{R}^2$. The *lower convex hull* is the lower boundary of this polygon, starting from $P_0 = (0, 0)$ to $P_K = (1, 1 + \lambda K)$.

Let the ordered sequence of vertices of the lower convex hull of $\mathcal{P}$ be $\{P_{v_0}, P_{v_1}, \ldots, P_{v_m}\}$, where $v_0 < v_1 < \cdots < v_m$ are indices from $\{0, \ldots, K\}$. By construction, we have $v_0 = 0$ and $v_m = K$, since $\Gamma_k$ are strictly increasing and $g_k$ are non-decreasing with $k$. For $i = 1, \ldots, m$, define the slope of the edge connecting the vertices $P_{v_{i-1}}$ and $P_{v_i}$ as $\eta_i := (g_{v_i} - g_{v_{i-1}})/(\Gamma_{v_i} - \Gamma_{v_{i-1}})$. To unify the analysis, we define $\eta_0 := 0$ and $\eta_{m+1} := +\infty$. The following theorem (with proof in Appendix B) characterizes the jumps and slopes of $\kappa$. Figure 1 shows an example of a lower convex hull of a point set $\mathcal{P}$ for a probability vector with $K = 10$.[6]

**Theorem 2.5** (Characterizing the optimal index function $\kappa$). *The range of $\kappa(\eta; \gamma)$ for $\eta \in [0, \infty)$ is precisely the set $\{v_0, v_1, \ldots, v_m\}$ of indices of the vertices of the lower convex hull. Moreover, the discontinuity points of $\eta \mapsto \kappa(\eta; \gamma)$ are the slopes $\eta_i$, $i = 1, \ldots, m$ of the edges of the vertices. Specifically,*

$$\kappa(\eta; \gamma) = \begin{cases} 0, & \text{for } \eta \in [0, \eta_1] \\ v_i, & \text{for } \eta \in (\eta_i, \eta_{i+1}], \ 1 \leqslant i \leqslant m - 1 \\ K, & \text{for } \eta \in (\eta_m, \infty). \end{cases}$$

**Computing the nonconformity score.** With Theorem 2.5, we can efficiently compute the nonconformity scores and the final prediction sets. The form $r(x, y_i) = \inf\{\eta \geqslant 0 : y_i \in S_{\eta, \hat{p}(\cdot|x)}\} = \inf\{\eta \geqslant 0 : \kappa(\eta; \hat{p}(\cdot|x)) \geqslant i\}$ is equivalent to finding *the smallest slope $\eta_j$ that leads to a prediction set of size $v_j \geqslant i$*. Intuitively, the slope $\eta$ represents the "price" per unit of coverage relative to the set-size penalty. Navigating the lower convex hull corresponds to finding the minimum price required to "purchase" enough coverage to include the target label $y_i$ in the set. To compute this, we can first find the vertices of the lower convex hull (which can be done with a standard approach, see Algorithm 2), and identify the correct slope; these can be performed in a single loop. Having an efficient algorithm to compute the nonconformity score is useful in a variety of conformal prediction

---

[6]Red points indicate the hull vertices, and $\eta_i$ denote the corresponding slopes. The nonconformity scores are $r(x, y_1) = \eta_1, r(x, y_2) = \cdots = r(x, y_7) = \eta_2, r(x, y_8) = \eta_3, r(x, y_9) = \eta_4$, and $r(x, y_{10}) = \eta_5$.

methods, such as split conformal prediction (Papadopoulos et al., 2002), cross-conformal prediction (Vovk, 2015), Mondrian conformal prediction (Vovk et al., 2005), etc. In this paper, we will focus on split conformal prediction, which is one of the most popular and widely applicable methods.

To run split conformal prediction given a set of $n$ calibration data points and a desired target coverage level $1 - \alpha$ in $[0, 1]$, we can compute $\hat{q}$, the $(1 - \alpha)(1 + 1/n)$-th quantile of the nonconformity scores over the calibration set; see Algorithm 3 in the Appendix.

**Coverage guarantees.** Naturally, the guarantees of conformal prediction are inherited here. Specifically, if our calibration and test data point are exchangeable, then we have that $P(Y_{n+1} \in \hat{C}(X_{n+1})) \geqslant 1 - \alpha$, where the randomness is taken jointly over the calibration and test data.

**Computing the prediction set.** Consider a new data point $x_{n+1}$ for which we aim to compute the prediction set $\hat{C}(x_{n+1})$. The range of $r(x_{n+1}, y_1), \cdots, r(x_{n+1}, y_K)$ is the set of the discontinuity points of $\kappa(\eta; \hat{p}(\cdot|x_{n+1}))$. Therefore, due to the monotonicity of $\kappa$ in $\eta$, we do not need to compute the score for each candidate label individually. Instead, we can directly search for the maximal slope along the lower convex hull that falls below the quantile $\hat{q}$; see Algorithm 1.

---

**Algorithm 1** SOCOP: Singleton-Optimized (Split) Conformal Prediction; with Singleton-Optimized Score

**Require:** Pre-trained model: $\hat{p}$, test point: $X_{n+1}$, penalty: $\lambda > 0$, $(1 - \alpha)(1 + 1/n)$-th quantile of calibration set nonconformity scores: $\hat{q}$.
**Ensure:** A prediction set $\hat{C}(X_{n+1})$ with coverage $1 - \alpha$.
1: Sort $\hat{p}(\cdot|X_{n+1})$ to get $\hat{p}_{\text{sorted}}(\cdot|X_{n+1})$ and associated labels $\text{idx}_{\text{sorted}, n+1}$
2: $(\mathcal{V}, \Gamma, g) \leftarrow$ Find lower convex hull using Algorithm 2 with input $(\hat{p}_{\text{sorted}}(\cdot|X_{n+1}), \lambda)$
3: $k_{\text{final}} \leftarrow 0$
4: **for** $j = 1$ **to** $|\mathcal{V}| - 1$ **do**
5: $\quad v_- \leftarrow \mathcal{V}[j-1]; \quad v_+ \leftarrow \mathcal{V}[j]; \quad \eta_j \leftarrow (g_{v_+} - g_{v_-}) / (\Gamma_{v_+} - \Gamma_{v_-})$
6: $\quad$ If $\eta_j \leqslant \hat{q}$ then $k_{\text{final}} \leftarrow v_+$, else break from for loop
7: **end for**
8: $\hat{C}(X_{n+1}) \leftarrow \{ \text{idx}_{\text{sorted}, n+1}[k] \; : \; 0 \leqslant k \leqslant k_{\text{final}} - 1 \}$
9: **return** $\hat{C}(X_{n+1})$

---

### 2.4 THE SCOPE OF OUR FRAMEWORK

In this section, we discuss certain important special cases and extensions of our methodology.

Our nonconformity score was derived starting from a linear combination of the singleton probability and the expected length. Therefore, it would be reassuring to know that our solution can indeed provably interpolate between the two by recovering them in certain limiting cases. In the next result, we show that this is indeed true and that our nonconformity score reduces to the corresponding nonconformity scores for these two cases. Recall below that we consider the labels to be sorted such that $\hat{p}(y_1|x) \geqslant \hat{p}(y_2|x) \geqslant \ldots$.

**Corollary 2.6** (Recovery of singleton objective optimization and least ambiguous sets). *(1) When $\lambda \to \infty$, the nonconformity scores have the limit $r_{\text{las}}(x, y_i) = 1/\hat{p}(y_i|x)$. The resulting split conformal prediction sets have the form $\{y \in \mathcal{Y} : \hat{p}(y|x_{n+1}) \geq c\}$, for some quantity $c$, recovering least ambiguous set-valued classifiers (Sadinle et al., 2019).*

*(2) When $\lambda = 0$, the nonconformity score becomes $r_{\text{singleton}}(x, y_i) = I(i \geq 2)\left(1 - \hat{p}(y_1 \mid x)\right)^{-1}$. The resulting split conformal prediction sets are either the top-1 label $\{y_1\}$ if $\hat{p}(y_1|x_{n+1}) \geqslant c$, for some quantity $c$, or the whole set $\mathcal{Y}$ otherwise.*

The proof of Corollary 2.6 is provided in Appendix B. The solution to the pure singleton objective has an intriguing structure. The prediction sets are *either the top-label or the full-label set*. This is intuitively reasonable: in the singleton objective, we are not paying any cost for the first label included in the set, so it makes sense to always include the most confident label. Moreover, we are paying full cost for any additional label included, and thus to ensure coverage, it is reasonable to include all labels into the prediction set.

However, this dichotomous behavior may not provide enough granularity in practice and may often output large prediction sets. This motivates our approach of taking a linear combination between the singleton and the length objectives. Our empirical results demonstrate that the nonconformity

scores derived from this linear combination offer a favorable trade-off, significantly reducing the probability of non-singletons compared, while only increasing the length by a little.

**Extension to** $P(\text{size} > k_0)$**.** Beyond controlling the probability of non-singletons, in some applications it might instead be more desirable to control the probability of sets larger than some other number, such as two, three, or ten. For instance, we might have two employees check one output each, and thus we might tolerate prediction sets of size two. Therefore, it is desirable to extend our framework to control the probability $\mathbb{P}_X[|C(X)| > k_0]$ of sets size larger than $k_0 \in \{1, \ldots, K-1\}$.

Fortunately, it turns out that our methods extend seamlessly to this case. We now seek to minimize $\mathbb{P}_X[|C(X)| > k_0] + \lambda \mathbb{E}_X[|C(X)|]$, subject to the same coverage constraint. The corresponding Lagrangian and separability arguments proceed similarly, with the difference that in the problem (4), $g_k$ in the cost function becomes $g_k = I(k > k_0) + \lambda k, k = 0, 1, \ldots, K$. The remaining steps are identical. The nonconformity score for the case $\lambda = 0$ from Corollary 2.6 becomes $r_{\text{top-k}}(x, y_i) = I(\{i > k_0\})/(1 - \sum_{j=1}^{k_0} \hat{p}(y_j \mid x))$. The corresponding sets consist of either the top $k_0$ indices or the full set.

## 3 EXPERIMENTS

In this section we report experiments comparing our `SOCOP` method with several prediction sets. The first one uses the probabilities output by the classifier directly, sorts them in decreasing order, and outputs the smallest set of classes whose predicted probabilities sum to at least $1 - \alpha$; we call this the `Plug-In` sets.[7] We also report results with split conformal prediction sets using a variety of nonconformity scores such as `RAPS` (Angelopoulos et al., 2021); `Pure Singleton` ($\lambda = 0$); `Least Ambiguous Sets` (Sadinle et al., 2019), corresponding to the nonconformity score[8] $(x, y) \mapsto 1 - \hat{p}(y|x)$, which recovers the case $\lambda \to \infty$ in our method. We additionally evaluate the `CPL` method proposed by Kiyani et al. (2024). This approach employs the same nonconformity score as `Least Ambiguous Sets`, but is conceptually different, as it replaces split conformal prediction with a training procedure to optimize prediction set length. Results for this method are reported in Table 17 in Appendix D.3.

For our proposed `SOCOP` method, the hyperparameter $\lambda$ is selected by aiming to find a "knee point" of the size-singleton probability curve on the tuning subset, as detailed in Section 3.1.1. The evaluation metrics we use are `Coverage`, `Average Size`, and `P(size>1)` [9].

### 3.1 IMAGE CLASSIFICATION ON IMAGENET

First, we consider image classification on the ImageNet-Val and ImageNet-V2 datasets, with several models, including `ResNet152-v2`, `EfficientNet-v2-l`, and `ViT-h-14`.

**Evaluation on ImageNet-Val.** In this experiment, we randomly sample three subsets of ImageNet-Val over 100 trials: one tuning subset of size 10K, one conformal calibration subset of size 20K and one evaluation subset of size 20K. For the `RAPS` baseline, we employ the hyperparameter tuning method from Algorithm 4 of Angelopoulos et al. (2021). The details of the hyperparameter grid are provided in Appendix E.

The averaged results for `ResNet152-v2` and `ViT-h-14`, along with standard errors, are reported in Table 1. Results for `EfficientNet-v2-l`, `ConvNeXt-base`, and `Swin-v2-b` are in Appendix D.1, and show similar trends. All methods achieve the target coverage of 0.95. Our method `SOCOP` outperforms `Plug-In` and `RAPS` in both `Average Size` and `P(size>1)`. Compared to `Least Ambiguous Sets`, `SOCOP` maintains a good balance: it produces sets nearly as small as `Least Ambiguous Sets` while significantly reducing the probability of non-singletons.

---

[7]This strategy is not theoretically guaranteed to attain the nominal level of coverage. However, it can be viewed as a reasonable empirically motivated baseline that practitioners might use by default.

[8]In Corollary 2.6, we wrote this non-conformity score as $1/\hat{p}(y|x)$; These are equivalent since any strictly monotone transformation of a nonconformity score induces the same prediction sets.

[9]We also evaluated the empty set rate, $P(|C(X)| = 0)$, across all experiments. We observed that empty sets occur in fewer than 0.01% of test instances for all methods, with the exception of RAPS (where they can be slightly larger but still insignificant, reaching $\approx 0.1\%$). Consequently, we omit this metric from the results as its impact is negligible.

Table 1: Performance on ImageNet-Val, for a Coverage of $1 - \alpha = 0.95$; Methods compared: `Plug-In`, `RAPS` (Angelopoulos et al., 2021), `Pure Singleton` ($\lambda = 0$), `Least Ambiguous Sets` ($\lambda = \infty$) (Sadinle et al., 2019; Kiyani et al., 2024) and our method `SOCOP`. Results are averages over 100 random splits. The smallest values in each column are highlighted in green, while all results worse than our method are highlighted in red. For our method `SOCOP`, the Avg Size and $P(\text{size} > 1)$ are highlighted in light green to facilitate comparison across models.

| Model | Method | Coverage | Avg Size | $P(\text{size} > 1)$ |
|---|---|---|---|---|
| ResNet152-v2 | Plug-In | $0.968 \pm 0.003$ | $44.955 \pm 5.558$ | $0.460 \pm 0.019$ |
| | RAPS | $0.950 \pm 0.002$ | $3.158 \pm 0.101$ | $0.603 \pm 0.154$ |
| | Pure Singleton | $0.949 \pm 0.002$ | $249.453 \pm 4.960$ | $0.249 \pm 0.005$ |
| | Least Ambiguous Sets | $0.950 \pm 0.002$ | $2.274 \pm 0.046$ | $0.466 \pm 0.007$ |
| | **SOCOP** (ours) | $0.950 \pm 0.002$ | $2.477 \pm 0.048$ | $0.370 \pm 0.006$ |
| ViT-h-14 | Plug-In | $0.976 \pm 0.001$ | $8.529 \pm 0.805$ | $0.356 \pm 0.008$ |
| | RAPS | $0.950 \pm 0.002$ | $1.380 \pm 0.020$ | $0.314 \pm 0.031$ |
| | Pure Singleton | $0.950 \pm 0.002$ | $136.219 \pm 4.980$ | $0.135 \pm 0.005$ |
| | Least Ambiguous Sets | $0.950 \pm 0.002$ | $1.291 \pm 0.011$ | $0.224 \pm 0.006$ |
| | **SOCOP** (ours) | $0.950 \pm 0.002$ | $1.356 \pm 0.017$ | $0.175 \pm 0.006$ |

**Evaluation on ImageNet-V2.** We apply the same evaluation pipeline to ImageNet-V2 (Recht et al., 2019), which is a more challenging test dataset. This dataset was constructed by re-collecting images with a new sampling pipeline, introducing a natural distribution shift that typically results in a significant drop in accuracy for models trained on the original ImageNet dataset. Since this is a smaller dataset, we randomly sample three subsets of Imagenet-Val over 100 trials: one tuning subset of size 1K, one conformal calibration subset of size 4K, and one evaluation subset of size 4K. Table 11 in Appendix D.2 reports the results for all five models. The empirical findings are consistent with those on ImageNet-Val. Notably, the advantages of `SOCOP` are even more pronounced on this more challenging dataset. The variance of the coverage is higher due to having less data.

### 3.1.1 EFFECT OF $\lambda$ AND HYPERPARAMETER TUNING

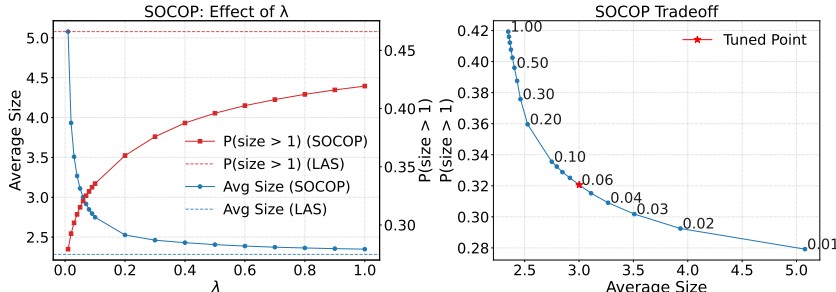

Figure 2: Visualizing the evaluation results of `ResNet152-v2` on ImageNet-Val from Table 6. LAS denotes `Least Ambiguous Sets`. Left: Average size and $P(\text{size} > 1)$ varying with $\lambda$; Right: visualization of (Average size, $P(\text{size} > 1)$), each point corresponding to a specific $\lambda$. Results corresponding to the hyperparameter $\lambda$ selected by the kneedle algorithm (Satopaa et al., 2011) are highlighted.

Next, we study the effect of the regularization parameter $\lambda$ on `Average Size` and `P(size>1)`. See Figure 2 for the trade-offs on the `ResNet152-v2` model evaluated over ImageNet-Val. Results are averaged over 100 random splits of Imagenet-Val, each of size 20K for calibration and 20K for evaluation. As $\lambda$ goes from 0 to $\infty$, the `Average Size` decreases from the level of the `Pure Singleton` ($\lambda = 0$) and converges to the `Least Ambiguous Sets` limit ($\lambda = \infty$); while `P(size>1)` follows an opposite trajectory.

The right panel of Figure 2 summarizes this trade-off by plotting `P(size>1)` against `Average Size`. In practice, one can choose $\lambda$ according to their own preference by drawing the tradeoff plot (the right panel of Figure 2) on their tuning dataset. For illustration, in our experiments from Section 3.1 we use the kneedle algorithm (Satopaa et al., 2011), which is a popular method for choosing points along a trade-off curve that come with favorable trade-offs. All five models exhibit the same pattern on both ImageNet-Val and ImageNet-V2, see Tables 6-16 in the Appendix.

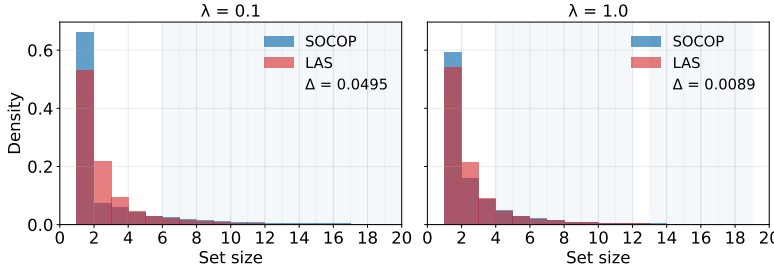

Figure 3: Set sizes produced with `ResNet152-v2` on ImageNet-Val. LAS denotes `Least Ambiguous Sets`. Bars indicate empirical probabilities of set sizes, and shaded bins mark non-singleton set sizes where `SOCOP` assigns higher mass. Reported $\Delta$ values denote the cumulative probability difference on shaded bins. The x-axis is truncated at 20 clarity.

We investigate the effect of regularization in more detail. For two values of $\lambda$, we collect the set sizes produced by `SOCOP` and `Least Ambiguous Sets`, and report their histograms in Figure 3. The figure shows that `SOCOP` yields more singleton sets and fewer small set sets (as desired), but produces slightly more sets with a large size sets (as expected due to the tradeoff). To quantify this shift toward larger sets, we calculate the cumulative excess probability mass $\Delta$ of `SOCOP` over `Least Ambiguous Sets` on non-singleton sizes, i.e., $\Delta := \sum_{i=2}^{K} I(f_i^{\text{SOCOP}} > f_i^{\text{LAS}})(f_i^{\text{SOCOP}} - f_i^{\text{LAS}})$, where $f_i^{\text{SOCOP}}, f_i^{\text{LAS}}$ are empirical frequencies of prediction set size $i$ for the two methods. We observe that this value is small, meaning that our method only leads to slightly more large sets.

### 3.1.2 ADAPTIVENESS ON IMAGENET

In this experiment, we evaluate the size-stratified coverage violation (SSCV) introduced by Angelopoulos et al. (2021) as a measure of adaptiveness and conditional coverage violation. Following Angelopoulos et al. (2021), we adopt the same set-size strata : 0-1, 2-3, 4- 10, 11-100, and 101-1000, and to maximize adaptiveness, we choose the hyperparameter $\lambda$ to minimize SSCV on the tuning set for `RAPS` and `SOCOP`. The details of the hyperparameter grid are provided in Appendix E. The results are reported in Table 2.

Table 2: Evaluation results for the SSCV metric on ImageNet-Val, with the same protocol as in Table 1.

| Method | Coverage | Avg Size | $P(\text{size} > 1)$ | SSCV |
|---|---|---|---|---|
| `ResNet152-v2` | | | | |
| `Plug-In` | $0.969 \pm 0.003$ | $47.362 \pm 7.138$ | $0.469 \pm 0.025$ | $0.046 \pm 0.001$ |
| `RAPS` | $0.950 \pm 0.002$ | $8.568 \pm 1.580$ | $0.448 \pm 0.012$ | $0.031 \pm 0.011$ |
| `Pure Singleton` | $0.950 \pm 0.002$ | $250.539 \pm 4.554$ | $0.250 \pm 0.005$ | $0.050 \pm 0.000$ |
| `Least Ambiguous Sets` | $0.950 \pm 0.002$ | $2.279 \pm 0.046$ | $0.467 \pm 0.007$ | $0.197 \pm 0.026$ |
| **SOCOP** (ours) | $0.950 \pm 0.002$ | $3.372 \pm 0.198$ | $0.304 \pm 0.008$ | $0.039 \pm 0.009$ |
| `ViT-h-14` | | | | |
| `Plug-In` | $0.976 \pm 0.001$ | $8.529 \pm 0.805$ | $0.356 \pm 0.008$ | $0.048 \pm 0.002$ |
| `RAPS` | $0.950 \pm 0.002$ | $7.652 \pm 2.259$ | $0.319 \pm 0.007$ | $0.047 \pm 0.003$ |
| `Pure Singleton` | $0.950 \pm 0.003$ | $136.219 \pm 4.980$ | $0.135 \pm 0.005$ | $0.050 \pm 0.000$ |
| `Least Ambiguous Sets` | $0.950 \pm 0.002$ | $1.291 \pm 0.011$ | $0.224 \pm 0.006$ | $0.126 \pm 0.119$ |
| **SOCOP** (ours) | $0.950 \pm 0.002$ | $1.519 \pm 0.068$ | $0.155 \pm 0.006$ | $0.041 \pm 0.016$ |

Our method `SOCOP` and `RAPS` achieve the smallest SSCV among the methods compared. However, the average size of `RAPS` increases drastically (from $\approx 3.2$ to $\approx 8.6$ for `Resnet152-v2` and from $\approx 1.4$ to $\approx 7.7$ for `ViT-h-14`), while our `SOCOP` method maintains a reasonably small average size and a significantly lower non-singleton probability, demonstrating that `SOCOP` can achieve adaptivity without sacrificing efficiency.

## 3.2 Image classification on TissueMNIST

We further evaluate on a medical image classification problem. We use the TissueMNIST dataset, a subset of MedMNIST (Yang et al., 2023), which contains microscopy images of human kidney cortex cells categorized into eight classes. We use a `ResNet-50(224)` model released by the dataset authors. We perform 100 random splits into 10K/15K/15K for tuning, calibration, and evaluation. The results are summarized in Table 3. As in previous experiments, we observe that our method can significantly reduce the non-singleton probability (by about 15%), while increasing the average size only slightly. This again validates the efficiency of our method.

Table 3: Evaluation results on TissueMNIST using `ResNet-50(224)` (Yang et al., 2023), with the same protocol as in Table 1.

| Method | Coverage | Avg Size | $P(\text{size} > 1)$ |
|---|---|---|---|
| Plug-In | $0.973 \pm 0.002$ | $3.294 \pm 0.030$ | $0.866 \pm 0.005$ |
| RAPS | $0.950 \pm 0.003$ | $2.844 \pm 0.031$ | $0.844 \pm 0.006$ |
| Pure Singleton | $0.950 \pm 0.003$ | $4.931 \pm 0.053$ | $0.562 \pm 0.008$ |
| Least Ambiguous Sets | $0.950 \pm 0.003$ | $2.647 \pm 0.028$ | $0.788 \pm 0.005$ |
| **SOCOP** (ours) | $0.950 \pm 0.003$ | $2.847 \pm 0.037$ | $0.638 \pm 0.009$ |

## 3.3 Multiple Choice Question Answering

We also evaluate on MMLU (Hendrycks et al., 2021), a multiple-choice question answering dataset. Following the same evaluation pipeline, we perform 100 random splits into 4K/5K/5K for tuning, calibration, and evaluation. We use Llama-3.1-8B-Instruct (Dubey et al., 2024), and following Kiyani et al. (2024), we input the fixed prompt: "*This is a 4-choice question that you should answer: {question}{choices} The correct answer to this question is:* ". We then extract the logits of the first output token corresponding to the answer options A, B, C, and D. Applying the softmax function yields probabilities over the four choices. The results are summarized in Table 4. As in previous experiments, we observe that our method can reduce the probability that the set size is greater than one by a significant amount (about 10%), while only increasing the average size by a negligible amount. This further reinforces that our approach provides a favorable trade-off between size and non-singleton probability.

Table 4: Evaluation on MMLU using `Llama-3.1-8B-Instruct`, with the same protocol as in Table 1.

| Method | Coverage | Avg Size | $P(\text{size} > 1)$ |
|---|---|---|---|
| Plug-In | $0.965 \pm 0.002$ | $2.648 \pm 0.013$ | $0.745 \pm 0.005$ |
| RAPS | $0.950 \pm 0.004$ | $2.601 \pm 0.032$ | $0.779 \pm 0.025$ |
| Pure Singleton | $0.950 \pm 0.004$ | $2.633 \pm 0.029$ | $0.544 \pm 0.010$ |
| Least Ambiguous Sets | $0.950 \pm 0.004$ | $2.426 \pm 0.030$ | $0.675 \pm 0.008$ |
| **SOCOP** (ours) | $0.950 \pm 0.004$ | $2.477 \pm 0.034$ | $0.587 \pm 0.016$ |

## 3.4 Discussion

SOCOP reframes efficiency in conformal classification around the goal of producing singletons, deriving a nonconformity score from a geometric analysis of a Lagrangian relaxation of the singleton objective. This yields an $O(K)$ per-instance algorithm, enabling split conformal sets that preserve marginal coverage while substantially increasing singleton frequency with minimal impact on average size. Empirically, over image classification and LLM multiple-choice benchmarks, this reduces non-singleton rates significantly relative to length-optimized baselines at near-identical set sizes; suggesting that our method could be broadly useful in practice.

In future work, it would be of interest to extend this method to more advanced conformal prediction methods, such as label-conditional or Mondrian conformal prediction (Vovk et al., 2005). Furthermore, a challenging but interesting theoretical direction is to design a nonconformity score intrinsically targeted for conditional coverage. This would likely entail retracing the derivation of SOCOP starting from a conditional-aware optimization objective, such as the one from Gibbs et al. (2025). Finally, while our current protocol uses a separate tuning set to maintain validity, future work could investigate data-dependent selection of $\lambda$ using the calibration set directly to improve data efficiency, while accounting for the resulting tuning bias (Zeng et al., 2025).

## REPRODUCIBILITY STATEMENT

All experimental details, including dataset information and evaluation protocols, are provided in Section 3 and appendix D. A GitHub repository, containing the implementation of SOCOP, baseline methods, and code to reproduce all experiments, is available at this repository. All theoretical results and assumptions are stated in Section 2.1, with complete proofs provided in Appendix B.

## ACKNOWLEDGMENTS

This work was partially supported by the NSF, ARO, AFOSR, ONR, and the Sloan Foundation.

## LLM USAGE

LLMs did not play a significant role in this work and were only used for grammar polishing in writing.

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

## A ADDITIONAL RELATED WORK

The non-parametric techniques which have been studied in conformal prediction belong to a much broader tradition of predictive inference in statistics which over the years have been developed both under parametric and non-parametric assumptions. See for instance Geisser (2017) and more recent works such as Bates et al. (2021); Park et al. (2022a;b); Sesia et al. (2023); Qiu et al. (2023); Si et al. (2024); Lee et al. (2025); Bashari et al. (2025); Joshi et al. (2025), which concern problems under a variety of assumptions.

Regarding the efficiency and optimality of conformal prediction, early work by Takeuchi in the 1970s—reviewed in Takeuchi (2020)—has established fundamental results, such as the fact that conformal prediction with a conformity score equal to a particular density $f$ is optimal—in terms of minimizing the expected length at the distribution with density $f$—among all methods of predictive inference that have marginal coverage over all distributions. Modern work has revisited optimality questions from a variety of different angles, as discussed in the main paper.

Conformal-type techniques have been developed to be used beyond standard classification and regression problems, for instance in sampling from large semantic spaces with generative AI models, see e.g., Horwitz & Hoshen (2022); Teneggi et al. (2023); Quach et al. (2024); Mohri & Hashimoto (2024); Chan et al. (2025), etc; and see Dobriban (2025) for a review. In our work, we provide an illustration for language model multiple-choice question answering, which becomes a conventional classification problem.

Recent frameworks (Liang et al., 2024; Yang & Kuchibhotla, 2025) optimize efficiency by *selecting* a nonconformity score from a pre-specified candidate set that minimizes a target loss. This differs fundamentally from our approach, which uses the loss to *derive* the score directly. Applying our composite loss within these selection frameworks faces two practical difficulties: (1) the resulting performance is strictly bounded by the pre-defined candidate pool; and (2) this would introduce the additional difficulty of selecting the hyperparameter $\lambda$, and their framework would have to be potentially extended to allow the selection of not just non-conformity scores but also loss functions.

## B AUXILIARY RESULTS AND PROOFS

### B.1 PROOF OF LEMMA 2.1

*Proof.* Clearly, $S_{0,\gamma} = \emptyset = \mathcal{F}_0$. Now let $\eta > 0$. If $S_{\eta,\gamma}$ is empty, then the claim holds; therefore, we only need to consider the case where $S_{\eta,\gamma}$ is non-empty. Assume $S_{\eta,\gamma}$ includes $y_{i_1}$ but not $y_{i_2}$ where $\gamma_{y_{i_2}} > \gamma_{y_{i_1}}$. Then we can construct $S' = (S_{\eta,\gamma} \setminus \{y_{i_1}\}) \cup \{y_{i_2}\}$ such that

$$\ell_\gamma(S';\eta) = \ell_\gamma(S_{\eta,\gamma};\eta) + \eta \sum_{y \in S_{\eta,\gamma}} \gamma_y - \eta \sum_{y \in S'} \gamma_y = \ell_\gamma(S_{\eta,\gamma};\eta) + \eta \left( \gamma_{y_{i_1}} - \gamma_{y_{i_2}} \right) < \ell(S_{\eta,\gamma};\eta),$$

which contradicts with the optimality of $S_{\eta,\gamma}$. Therefore, $S_{\eta,\gamma}$ must be the set of top-$j$ labels for some $j$ that depends on $\eta$ and $\gamma$. $\qquad\square$

## B.2 PROOF OF LEMMA 2.2

*Proof.* For any set $S \in [K]$, we define $g(S) = I(|S| > 1) + \lambda |S|$ and $\Gamma(S) = \sum_{y \in S} \gamma_y$. We will first prove that for $\eta_1 < \eta_2$, $\Gamma(S_{\eta_1,\gamma}) \leq \Gamma(S_{\eta_2,\gamma})$. Then by Lemma 2.1, we must have $S_{\eta_1,\gamma} \subseteq S_{\eta_2,\gamma}$.

By the optimality of each set, we have

- $\ell\big(S_{\eta_1,\gamma}, \eta_1\big) \leqslant \ell\big(S_{\eta_2,\gamma}, \eta_1\big)$, i.e., $g\big(S_{\eta_1,\gamma}\big) - \eta_1 \Gamma\big(S_{\eta_1,\gamma}\big) \leqslant g\big(S_{\eta_2,\gamma}\big) - \eta_1 \Gamma\big(S_{\eta_2,\gamma}\big)$,

- $\ell\big(S_{\eta_2,\gamma}, \eta_2\big) \leqslant \ell\big(S_{\eta_1,\gamma}, \eta_2\big)$, i.e., $g\big(S_{\eta_2,\gamma}\big) - \eta_2 \Gamma\big(S_{\eta_2,\gamma}\big) \leqslant g\big(S_{\eta_1,\gamma}\big) - \eta_2 \Gamma\big(S_{\eta_1,\gamma}\big)$.

Combining these inequalities gives

$$\eta_2\big(\Gamma(S_{\eta_1,\gamma}) - \Gamma(S_{\eta_2,\gamma})\big) \leqslant g(S_{\eta_1,\gamma}) - g(S_{\eta_2,\gamma}) \leqslant \eta_1\big(\Gamma(S_{\eta_1,\gamma}) - \Gamma(S_{\eta_2,\gamma})\big).$$

Since $\eta_2 - \eta_1 > 0$, we have $\Gamma(S_{\eta_1,\gamma}) - \Gamma(S_{\eta_2,\gamma}) \leqslant 0$. By Lemma 2.1, for any $\eta \geqslant 0$ and $\gamma \in \Delta_{K-1}$, $S_{\eta,\gamma}$ is the set of top-$j$ labels for some $j$. Then $\Gamma(S_{\eta_1,\gamma}) \leq \Gamma(S_{\eta_2,\gamma})$ implies that $S_{\eta_1,\gamma} \subseteq S_{\eta_2,\gamma}$. $\quad\square$

## B.3 PROOF OF THEOREM 2.5:

To simplify notation, let us denote $\ell_\gamma^{(k)}(\eta) := \ell_\gamma(\mathcal{F}_k; \eta) = g_k - \eta\Gamma_k$. Then the minimization problem in (4) can be rewritten as $\kappa(\eta; \gamma) = \arg\min_k \ell_\gamma^{(k)}(\eta)$. The optimal value of the objective is given by

$$\ell_\gamma^*(\eta) := \min_{0 \leqslant k \leqslant K} \ell_\gamma^{(k)}(\eta).$$

This problem can be analyzed from two viewpoints, see also Figure 4.

- **Dual Space** $(\eta, \ell)$**:** For each $k \in \{0, \ldots, K\}$, we can view $\ell_\gamma^{(k)}(\eta) = g_k - \eta\Gamma_k$ as a linear function of $\eta$ for $(\eta, \ell) \in \mathbb{R}^2$. For a fixed value $\eta$, the optimal value $\ell_\gamma^*(\eta)$ corresponds to finding the lowest point among the intersections of the $K + 1$ lines with the vertical line $\ell = \eta$. As shown in the left plot of Figure 4, the function $\ell_\gamma^*$ forms the lower envelope of this family of $K + 1$ lines. The vertices of this lower envelope correspond to the values of $\eta$ where the optimal index $\kappa(\eta; \gamma)$ transitions from one value to another.

- **Primal Space** $(\Gamma, g)$**:** Since $g_k = \eta\Gamma_k + \ell_\gamma^{(k)}(\eta)$, $\ell_\gamma^{(k)}(\eta)$ can be viewed as the intercept of a line with slope $\eta$ that passes through the point $P_k = (\Gamma_k, g_k)$. For a fixed $\eta$, in the space of $(\Gamma, g)$, $\{g = \eta\Gamma + \ell, \ell \in \mathbb{R}\}$ is a family of parallel lines. Minimizing $\ell_\gamma^{(k)}(\eta)$ over $k$ amounts to finding the first point in $\{P_0, \ldots P_K\}$ that is "hit" by such a line as the intercept $\ell$ raises from $-\infty$.

Mathematically, the duality between these two perspectives can be formalized using convex conjugacy. Define a primal function $\phi : [0, 1] \to [0, \infty]$ based on the point set $\mathcal{P} = \{P_0, \ldots, P_K\}$:

$$\phi(\Gamma) = \begin{cases} g_k & \text{if } \Gamma = \Gamma_k, \ 0 \leq k \leqslant K \\ +\infty & \text{otherwise.} \end{cases}$$

The convex conjugate (see e.g., Rockafellar, 1997) of $\phi$ is

$$\phi^*(\eta) = \sup_{\Gamma \in \mathbb{R}}\{\eta\Gamma - \phi(\Gamma)\} = \max_{0 \leq k \leqslant K}\{\eta\Gamma_k - g_k\} = -\ell_\gamma^*(\eta). \tag{6}$$

Furthermore, the biconjugate of $\phi$, defined as the conjugate of $\phi^*$ is

$$\phi^{**}(\Gamma) = \sup_{\eta \in \mathbb{R}}\{\eta\Gamma - \phi^*(\eta)\} = \sup_{\eta \in \mathbb{R}}\{\eta\Gamma + \ell_\gamma^*(\eta)\}. \tag{7}$$

By the Fenchel-Moreau-Rockafellar theorem (see e.g. Theorem 3.2.2 in Correa et al. (2023)), $\phi^{**}$ is the closed convex hull of the original function $\phi$: $\phi^{**} = \overline{\mathrm{co}}(\phi)$, where $\overline{\mathrm{co}}(\phi)$ denotes the closed convex hull of $\phi$. Thus, it suffices to characterize $\overline{\mathrm{co}}(\phi)$, which we will do through its epigraph.

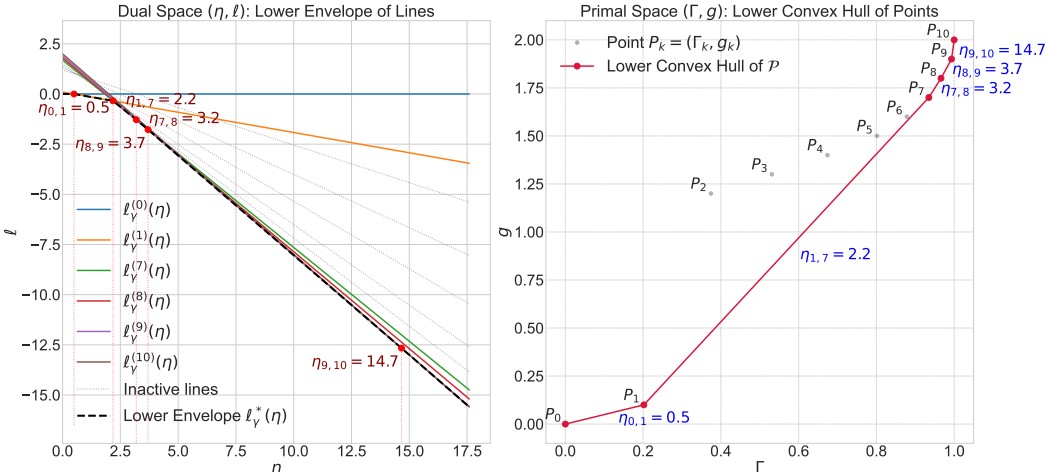

Figure 4: Primal and dual views, for $\lambda = 0.1$ and the example probability vector $\gamma$ is $[0.202, 0.172, 0.157, 0.143, 0.127, 0.077, 0.057, 0.031, 0.027, 0.007]$.

Let $\mathrm{epi}(\phi)$ denote the epigraph of $\phi$, defined as $\mathrm{epi}(\phi) = \bigcup_{k=1}^{K} \{(\Gamma_k, \omega) \mid \omega \geq g_k\}$. Using that $\mathrm{epi}(\overline{\mathrm{co}}(\phi)) = \overline{\mathrm{co}}(\mathrm{epi}(\phi))$, any point $(\Gamma, \omega) \in \overline{\mathrm{co}}(\phi)$ can be expressed as:

$$(\Gamma, \omega) = \sum_{k=1}^{K} \beta_k (\Gamma_k, \omega_k) \text{ for some } \beta_k \geq 0, \sum \beta_k = 1, \text{ and } (\Gamma_k, \omega_k) \in \mathrm{epi}(\phi).$$

Since $\omega = \sum_{k=1}^{K} \beta_k \omega_k \geq \sum_{k=1}^{K} \beta_k g_k$, and the minimum is attained, we know that

$$\phi^{**}(\Gamma) = \inf \left\{ \sum_{k=1}^{K} \beta_k g_k \mid \Gamma = \sum_{k=1}^{K} \beta_k \Gamma_k, \beta_k \geq 0, \sum \beta_k = 1 \right\} \tag{8}$$

which is precisely the lower convex hull of the point set $\mathcal{P}$.

We now continue with the proof of Theorem 2.5.

**Lemma B.1.** *An index $k$ is in the set of optimal solutions for some $\eta_0$ (i.e., $\ell_\gamma^{(k)}(\eta_0) = \ell_\gamma^*(\eta_0)$) if and only if its corresponding point $P_k$ lies on the graph of the lower convex hull function $\phi^{**}(\Gamma)$ (i.e., $g_k = \phi^{**}(\Gamma_k)$).*

*Proof.* ( $\implies$ ) Assume $\ell_\gamma^{(k)}(\eta_0) = \ell_\gamma^*(\eta_0)$. By definition, this implies $g_k - \eta_0 \Gamma_k = \ell_\gamma^*(\eta_0)$. From the biconjugate (7), $\phi^{**}(\Gamma_k) = \sup_\eta \{\Gamma_k \eta + \ell_\gamma^*(\eta)\} \geq \Gamma_k \eta_0 + \ell_\gamma^*(\eta_0) = g_k$. Since $\phi^{**}$ is the lower convex hull function of $\mathcal{P}$, we must also have $\phi^{**}(\Gamma_k) \leq g_k$. Therefore, $g_k = \phi^{**}(\Gamma_k)$.

( $\impliedby$ ) Assume that the point $P_k = (\Gamma_k, g_k)$ lies on the graph of the lower convex hull, i.e., $g_k = \phi^{**}(\Gamma_k)$. By the Supporting Hyperplane Theorem (Rockafellar, 1997), $P_k$ being on the lower boundary of convex hull implies that there exists a supporting line to the function $\phi^{**}$ at the point $\Gamma = \Gamma_k$. Let the slope of this supporting line be $\eta_k$. Then for all $\Gamma$ in the domain, we have $\phi^{**}(\Gamma) \geq \phi^{**}(\Gamma_k) + \eta_k(\Gamma - \Gamma_k)$.

For any $j \in \{0, \ldots, K\}$, $P_j = (\Gamma_j, g_j)$ must lie on or above the lower convex hull, i.e., $g_j \geq \phi^{**}(\Gamma_j)$. Applying this to the inequality above for $\Gamma = \Gamma_j$, we find:

$$g_j \geq \phi^{**}(\Gamma_j) \geq \phi^{**}(\Gamma_k) + \eta_k(\Gamma_j - \Gamma_k).$$

By our initial assumption $\phi^{**}(\Gamma_k) = g_k$, then we have $g_j \geq g_k + \eta_k(\Gamma_j - \Gamma_k)$. This inequality holds for all $j \in \{0, \ldots, K\}$. Thus, $g_j - \eta_k \Gamma_j \geq g_k - \eta_k \Gamma_k$, that is, $\ell_\gamma^{(j)}(\eta_k) \geq \ell_\gamma^{(k)}(\eta_k)$ for all $j \in \{0, \ldots, K\}$. Therefore, $\ell_\gamma^{(k)}(\eta_k) = \ell_\gamma^*(\eta_k)$, i.e., $k$ is an optimal index for $\eta = \eta_k$. $\qquad \square$

Recall that the vertices of the lower convex hull of $\mathcal{P}$ are $\{P_{v_0}, P_{v_1}, \ldots, P_{v_m}\}$, where $0 = v_0 < v_1 < \cdots < v_m = K$ are indices from $\{0, \ldots, K\}$. Recall from Section 2.3 that for $i = 1, \ldots, m$, the slope of the edge connecting vertex $P_{v_{i-1}}$ and $P_{v_i}$ is defined as $\eta_i := \frac{g_{v_i} - g_{v_{i-1}}}{\Gamma_{v_i} - \Gamma_{v_{i-1}}}$ where we define $\eta_0 := 0$ and $\eta_{m+1} := +\infty$. From the definition of convexity, it follows that these slopes are strictly increasing: $0 < \eta_1 < \eta_2 < \cdots < \eta_m < \infty$. Our next result is the following:

**Lemma B.2** (Unique Optimality on Vertex Intervals). *For any $\eta \in (\eta_i, \eta_{i+1})$, we have $\ell_\gamma^{(v_i)}(\eta) < \ell_\gamma^{(k)}(\eta)$ for all $k \neq v_i$.*

*Proof.* Let $\eta \in (\eta_i, \eta_{i+1})$ for a given $i \in \{0, 1, \ldots, m - 1\}$. We need to show that $g_k - g_{v_i} > \eta(\Gamma_k - \Gamma_{v_i})$ for any $k \neq v_i$. By Lemma B.1, any point not on the lower convex hull cannot be optimal, so it suffices to check this for other vertices $P_{v_j}$ where $j \neq i$.

**Case 1:** $j > i$ (i.e., $\Gamma_{v_j} > \Gamma_{v_i}$). Since $\phi^{**}$ is convex, for any $j > i$, we have

$$\eta_{i+1} = \frac{g_{v_{i+1}} - g_{v_i}}{\Gamma_{v_{i+1}} - \Gamma_{v_i}} \leqslant \frac{g_{v_j} - g_{v_i}}{\Gamma_{v_j} - \Gamma_{v_i}}.$$

By our choice of $\eta$, we have $\eta < \eta_{i+1}$. Combining these gives $\eta < \frac{g_{v_j} - g_{v_i}}{\Gamma_{v_j} - \Gamma_{v_i}}$. As $\Gamma_{v_j} - \Gamma_{v_i} > 0$, we have $\eta(\Gamma_{v_j} - \Gamma_{v_i}) < g_{v_j} - g_{v_i}$, so that $\ell_\gamma^{(v_i)}(\eta) < \ell_\gamma^{(v_j)}(\eta)$.

**Case 2:** $j < i$ (i.e., $\Gamma_{v_j} < \Gamma_{v_i}$). Similarly, for any $j < i$, by the convexity of $\phi^{**}$, we have:

$$\frac{g_{v_i} - g_{v_j}}{\Gamma_{v_i} - \Gamma_{v_j}} \leqslant \frac{g_{v_i} - g_{v_{i-1}}}{\Gamma_{v_i} - \Gamma_{v_{i-1}}} = \eta_i,$$

which implies $\ell_\gamma^{(v_i)}(\eta) < \ell_\gamma^{(v_j)}(\eta)$ as above.

This finishes the proof. $\qquad\square$

*Proof of Theorem 2.5.* From Lemma B.2, for any $\eta$ in the open interval $(\eta_i, \eta_{i+1})$, the unique minimizer is $v_i$, so $\kappa(\eta; \gamma) = v_i$. At the boundary points $\eta = \eta_i$ for $i \in \{1, \ldots, m\}$, we have $\ell_\gamma^{(v_{i-1})}(\eta_i) = \ell_\gamma^{(v_i)}(\eta_i)$ by definition. The proofs in Lemma B.2 show that for any other vertex $v_j$, $\ell_\gamma^{(v_j)}(\eta_i)$ is strictly greater. Thus, the set of optimal indices is $\{v_{i-1}, v_i\}$. By the tie-breaking rule[10], we have $\kappa(\eta_i; \gamma) = v_{i-1}$.

Combining these observations, for $i = 1, \ldots, m - 1$, and for any $\eta \in (\eta_i, \eta_{i+1}]$, the optimal index is $\kappa(\eta; \gamma) = v_i$. It is clear that $\kappa(0; \gamma) = 0$. Therefore, for $\eta \in [0, \eta_1]$, $\kappa(\eta; \gamma) = 0$. By Lemma B.2, for $\eta \in (\eta_m, \eta_{m+1}) = (\eta_m, \infty)$, $\kappa(\eta; \gamma) = v_m$. This finishes the proof. $\qquad\square$

### B.4 PROOF OF COROLLARY 2.6

Let $\gamma = \hat{p}(\cdot|x)$ be the probability prediction from some pre-trained model. We order the probabilities such that $\hat{p}(y_1|x) \geq \hat{p}(y_2|x) \geq \hat{p}(y_K|x)$.

(1) When dividing the Lagrangian (1) by $\lambda$, the problem remains equivalent by changing variables from $\eta$ to $\tilde{\eta} := \eta/\lambda$. The resulting $\kappa(\tilde{\eta}; \hat{p}(\cdot|x))$ becomes:

$$\kappa(\tilde{\eta}; \hat{p}(\cdot|x)) := \arg\min_{0 \leq k \leq K} \left\{ \frac{I(k > 1)}{\lambda} + k - \tilde{\eta} \cdot \sum_{i=1}^k \hat{p}(y_i|x) \right\}.$$

The corresponding set of points $\mathcal{P}$ becomes $P_k = \left( \sum_{i=1}^k \hat{p}(y_i|x), \frac{I(k>1)}{\lambda} + k \right)$. We first consider the simpler case where $\hat{p}(y_1|x) > \hat{p}(y_2|x) > \cdots > \hat{p}(y_K|x)$. When $\lambda \to \infty$, $P_k \longrightarrow \left( \sum_{i=1}^k \hat{p}(y_i|x), k \right) := \tilde{P}_k$. The slope between two consecutive points is: $1/\hat{p}(y_{k+1}|x)$, which is

---

[10]When there are multiple solutions, we choose any set that has minimal size, which corresponds to choosing the smallest index for $\kappa(\eta; \gamma)$

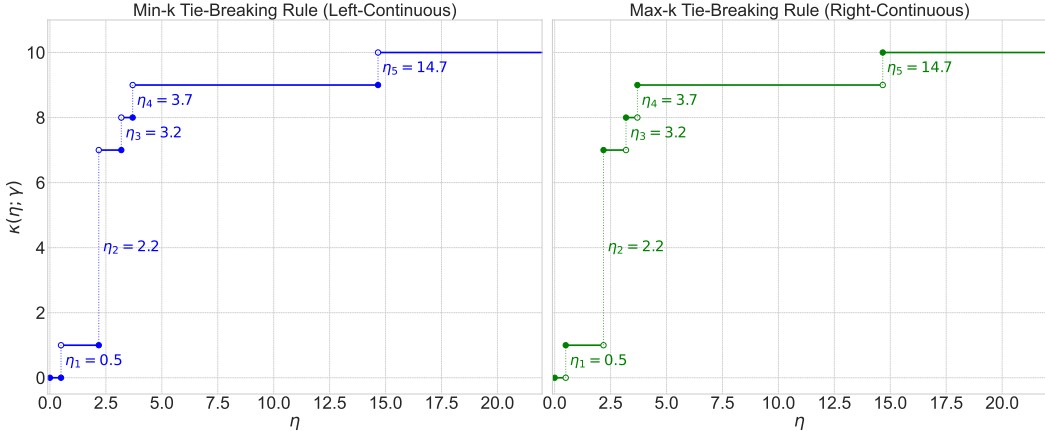

Figure 5: Optimal Set Size $\kappa(\eta; \gamma)$ with the same parameters as Figure 4. The tie-breaking rule does not affect the value of nonconformity score.

strictly increasing in $k$. Hence, every point $\tilde{P}_k$ is a vertex of the lower convex hull. Thus $\kappa(\tilde{\eta}; \hat{p}(\cdot|x))$ becomes:

$$\kappa(\tilde{\eta}; \hat{p}(\cdot|x)) = \begin{cases} 0, & \text{for } \eta \in \left[0, \frac{1}{\hat{p}(y_1|x)}\right] \\ i, & \text{for } \eta \in \left(\frac{1}{\hat{p}(y_i|x)}, \frac{1}{\hat{p}(y_{i+1}|x)}\right], 1 \leqslant i \leqslant K-1 \\ K, & \text{for } \eta \in \left(\frac{1}{\hat{p}(y_K|x)}, \infty\right). \end{cases}$$

Therefore, the nonconformity score is $r_{\text{las}}(x, y_i) = 1/\hat{p}(y_i|x)$.

Now, suppose there is a tie, e.g., $\hat{p}(y_k \mid x) = \hat{p}(y_{k+1} \mid x) = \cdots = \hat{p}(y_{k+m} \mid x)$ for some $k \geq 1, m \geq 1$. Then, the points $\tilde{P}_{k-1}, \tilde{P}_k, \ldots, \tilde{P}_{k+m}$ are collinear, with vertices $\tilde{P}_{k-1}$ and $\tilde{P}_{k+m}$ and slope $\frac{1}{\hat{p}(y_k|x)}$. The function $\kappa(\cdot; \hat{p}(\cdot|x))$ exhibits a single jump from $k-1$ to $k+m$ as $\tilde{\eta}$ crosses this slope value. For any label $y_i$ with $k \leq i \leq k+m$, the nonconformity score is

$$r_{\text{las}}(x, y_i) = \inf\{\tilde{\eta} \geq 0 : \kappa(\tilde{\eta}; \hat{p}(\cdot|x)) \geq i\} = 1/\hat{p}(y_k \mid x) = 1/\hat{p}(y_i \mid x),$$

as desired.

(2) When $\lambda = 0$, the set of points $\mathcal{P} = \{P_0, \ldots, P_K\}$ becomes

$$P_0 = (0, 0), \ P_1 = (\hat{p}(y_1 \mid x), 0), \ P_k = \left(\sum_{i=1}^k \hat{p}(y_i|x), 1\right) \ (k \geqslant 2), \ P_K = (1, 1).$$

As $\Gamma_k$ is strictly increasing in $k$, the vertices of the lower convex hull are $\{P_0, P_1, P_K\}$. The corresponding slopes are $\eta_1 = \frac{0-0}{\hat{p}(y_1|x)-0} = 0$ and

$$\eta_2 = \frac{g_K - g_1}{\sum_{i=1}^K \hat{p}(y_i|x) - \hat{p}(y_1|x)} = \frac{1}{1 - \hat{p}(y_1 \mid x)}.$$

Hence, $\kappa(\eta; \hat{p}(\cdot|x))$ becomes:

$$\kappa(\eta; \hat{p}(\cdot|x)) := \begin{cases} 0, & \text{for } \eta = 0 \\ 1, & \text{for } \eta \in \left(0, \frac{1}{1 - \hat{p}(y_1 \mid x)}\right], \\ K, & \text{for } \eta \in \left(\frac{1}{1 - \hat{p}(y_1 \mid x)}, \infty\right). \end{cases}$$

Therefore, by definition of the nonconformity score $r(x, y_i) = \inf\{\eta \geqslant 0 : \kappa(\eta; \hat{p}(\cdot|x)) \geqslant i\}$, we have $r_{\text{singleton}}(x, y_i) = I(i \geq 2)\left(1 - \hat{p}(y_1 \mid x)\right)^{-1}$. $\qquad \square$

## C  ADDITIONAL ALGORITHMS

We leverage the monotone chain algorithm (Andrew, 1979; o'Rourke, 1998) to find the vertices of lower convex hull, as detailed in Algorithm 2.

---

**Algorithm 2** Compute Lower Convex Hull via Monotone Chain Algorithm

---

**Require:** Sorted probability vector $p$, penalty $\lambda > 0$.
**Ensure:** A tuple $(\mathcal{V}, \Gamma, g)$ where $\mathcal{V}$ is the list of vertex indices of lower convex hull, $\Gamma$ are cumulative sums, and $g$ are objective values.
  1: Compute cumulative sums $\Gamma_k \leftarrow \sum_{j=1}^{k} p_j$ for $k = 0, \ldots, K$            $\triangleright S_0 = 0$
  2: Compute objective values $g_k \leftarrow I(k > 1) + \lambda k$ for $k = 0, \ldots, K$
  3: Define CrossProduct$(j, i, k; \Gamma, g) = (\Gamma_i - \Gamma_j)(g_k - g_i) - (g_i - g_j)(\Gamma_k - \Gamma_i)$.
  4: Initialize an empty list of indices $\mathcal{V}$.
  5: **for** $k = 0$ **to** $K$ **do**                                                           $\triangleright$ Monotone Chain
  6:     **while** $|\mathcal{V}| \geqslant 2$ **and** CrossProduct$(\mathcal{V}[-2], \mathcal{V}[-1], k; \Gamma, g) \leqslant 0$ **do**     $\triangleright$ Last two points
  7:         Remove the last index from $\mathcal{V}$.
  8:     **end while**
  9:     Append index $k$ to $\mathcal{V}$.
 10: **end for**
 11: **return** $(\mathcal{V}, \Gamma, g)$.

---

For calibration, we adopt standard split conformal prediction, see Algorithm 3.

---

**Algorithm 3** SOCOP Conformal Calibration

---

**Require:** Pre-trained model $\hat{p}$, calibration data $\{(X_i, Y_i)\}_{i=1}^{n}$, level $\alpha \in (0, 1)$, penalty $\lambda > 0$.
**Ensure:** Calibrated threshold $\hat{q}$.
  1: **for** $i = 1$ **to** $n$ **do**
  2:     Sort $\hat{p}(\cdot|X_i)$ to get $\hat{p}_{\text{sorted}}(\cdot|X_i)$
  3:     Let $i_{\text{rank}}$ be the 1-based rank of the true label $Y_i$
  4:     $(\mathcal{V}, \Gamma, g) \leftarrow$ Algorithm 2$(\hat{p}_{\text{sorted}}(\cdot|X_i), \lambda)$
  5:     Find the smallest index $j \in \{1, \ldots, |\mathcal{V}| - 1\}$ such that $\mathcal{V}[j] \geqslant i_{\text{rank}}$
  6:     $v_- \leftarrow \mathcal{V}[j-1]$;    $v_+ \leftarrow \mathcal{V}[j]$
  7:     $r_i \leftarrow \left(g_{v_+} - g_{v_-}\right) / \left(\Gamma_{v_+} - \Gamma_{v_-}\right)$
  8: **end for**
  9: $\hat{q} \leftarrow$ the $\lceil (1 - \alpha)(1 + n) \rceil$ largest value in $\{r_i\}_{i=1}^{n}$
 10: **return** $\hat{q}$

---

## D  ADDITIONAL EXPERIMENTS RESULTS

### D.1  IMAGENET-VAL

Results for `EfficientNet-v2-l`, `ConvNeXt-base`, and `Swin-v2-b` on the ImageNet-Val dataset are reported in Table 5.

For this dataset, the effect of $\lambda$ on our `SOCOP` across all five models are reported in Table 6-10, respectively.

### D.2  IMAGENET-V2

Results for all five models on the ImageNet-V2 dataset are reported in Table 11.

For this dataset, the effect of $\lambda$ on our `SOCOP` across all five models are reported in Table 12-16, respectively.

Table 5: Continuation of the results in Table 1 with the same protocol used.

| Model | Method | Coverage | Avg Size | $P(\text{size} > 1)$ |
|---|---|---|---|---|
| EfficientNet-v2-l | Plug-In | $0.970 \pm 0.002$ | $16.606 \pm 2.023$ | $0.401 \pm 0.013$ |
| | RAPS | $0.950 \pm 0.002$ | $1.909 \pm 0.077$ | $0.769 \pm 0.076$ |
| | Pure Singleton | $0.950 \pm 0.002$ | $188.942 \pm 4.836$ | $0.188 \pm 0.005$ |
| | Least Ambiguous Sets | $0.950 \pm 0.002$ | $1.542 \pm 0.018$ | $0.329 \pm 0.007$ |
| | **SOCOP** (ours) | $0.950 \pm 0.002$ | $1.659 \pm 0.023$ | $0.262 \pm 0.006$ |
| ConvNeXt-base | Plug-In | $0.967 \pm 0.003$ | $27.137 \pm 5.790$ | $0.444 \pm 0.030$ |
| | RAPS | $0.950 \pm 0.003$ | $2.546 \pm 0.096$ | $0.843 \pm 0.234$ |
| | Pure Singleton | $0.950 \pm 0.002$ | $226.991 \pm 4.935$ | $0.226 \pm 0.005$ |
| | Least Ambiguous Sets | $0.950 \pm 0.002$ | $1.897 \pm 0.034$ | $0.398 \pm 0.007$ |
| | **SOCOP** (ours) | $0.950 \pm 0.002$ | $2.086 \pm 0.046$ | $0.316 \pm 0.007$ |
| Swin-v2-b | Plug-In | $0.968 \pm 0.003$ | $19.646 \pm 3.701$ | $0.423 \pm 0.023$ |
| | RAPS | $0.950 \pm 0.002$ | $2.314 \pm 0.062$ | $0.477 \pm 0.121$ |
| | Pure Singleton | $0.950 \pm 0.002$ | $225.685 \pm 4.909$ | $0.225 \pm 0.005$ |
| | Least Ambiguous Sets | $0.950 \pm 0.002$ | $1.881 \pm 0.033$ | $0.396 \pm 0.007$ |
| | **SOCOP** (ours) | $0.950 \pm 0.002$ | $2.068 \pm 0.038$ | $0.316 \pm 0.007$ |

Table 6: Performance of ResNet152-v2 on ImageNet-Val with different $\lambda$ values ($\alpha = 0.05$). Results are averaged over 100 data splits.

| $\lambda$ | Method | Coverage | Avg Size | $P(\text{size} > 1)$ |
|---|---|---|---|---|
| 0 | Pure Singleton | $0.949 \pm 0.002$ | $249.453 \pm 4.960$ | $0.249 \pm 0.005$ |
| 0.01 | SOCOP (ours) | $0.950 \pm 0.002$ | $5.078 \pm 0.200$ | $0.279 \pm 0.005$ |
| 0.02 | SOCOP (ours) | $0.950 \pm 0.002$ | $3.932 \pm 0.125$ | $0.293 \pm 0.005$ |
| 0.03 | SOCOP (ours) | $0.950 \pm 0.002$ | $3.508 \pm 0.103$ | $0.302 \pm 0.006$ |
| 0.04 | SOCOP (ours) | $0.950 \pm 0.002$ | $3.267 \pm 0.104$ | $0.309 \pm 0.006$ |
| 0.05 | SOCOP (ours) | $0.950 \pm 0.002$ | $3.110 \pm 0.092$ | $0.315 \pm 0.006$ |
| 0.06 | SOCOP (ours) | $0.950 \pm 0.002$ | $3.002 \pm 0.081$ | $0.321 \pm 0.006$ |
| 0.07 | SOCOP (ours) | $0.950 \pm 0.002$ | $2.916 \pm 0.074$ | $0.325 \pm 0.006$ |
| 0.08 | SOCOP (ours) | $0.950 \pm 0.002$ | $2.847 \pm 0.071$ | $0.329 \pm 0.006$ |
| 0.09 | SOCOP (ours) | $0.950 \pm 0.002$ | $2.795 \pm 0.069$ | $0.332 \pm 0.006$ |
| 0.10 | SOCOP (ours) | $0.950 \pm 0.002$ | $2.749 \pm 0.068$ | $0.336 \pm 0.006$ |
| 0.20 | SOCOP (ours) | $0.950 \pm 0.002$ | $2.527 \pm 0.061$ | $0.360 \pm 0.006$ |
| 0.30 | SOCOP (ours) | $0.949 \pm 0.002$ | $2.461 \pm 0.058$ | $0.376 \pm 0.006$ |
| 0.40 | SOCOP (ours) | $0.950 \pm 0.002$ | $2.430 \pm 0.059$ | $0.388 \pm 0.007$ |
| 0.50 | SOCOP (ours) | $0.950 \pm 0.002$ | $2.406 \pm 0.054$ | $0.396 \pm 0.006$ |
| 0.60 | SOCOP (ours) | $0.950 \pm 0.002$ | $2.388 \pm 0.051$ | $0.403 \pm 0.006$ |
| 0.70 | SOCOP (ours) | $0.950 \pm 0.002$ | $2.373 \pm 0.051$ | $0.408 \pm 0.006$ |
| 0.80 | SOCOP (ours) | $0.950 \pm 0.002$ | $2.364 \pm 0.051$ | $0.412 \pm 0.006$ |
| 0.90 | SOCOP (ours) | $0.950 \pm 0.002$ | $2.355 \pm 0.053$ | $0.416 \pm 0.007$ |
| 1.00 | SOCOP (ours) | $0.950 \pm 0.002$ | $2.349 \pm 0.054$ | $0.419 \pm 0.007$ |
| $\infty$ | Least Ambiguous Sets | $0.950 \pm 0.002$ | $2.274 \pm 0.046$ | $0.466 \pm 0.007$ |

Table 7: Performance of EfficientNet-v2-l on ImageNet-Val with different $\lambda$ values ($\alpha = 0.05$). Results are averaged over 100 data splits.

| $\lambda$ | Method | Coverage | Avg Size | $P(\text{size} > 1)$ |
|---|---|---|---|---|
| 0 | Pure Singleton | $0.950 \pm 0.002$ | $188.942 \pm 4.836$ | $0.188 \pm 0.005$ |
| 0.01 | SOCOP (ours) | $0.950 \pm 0.002$ | $2.873 \pm 0.091$ | $0.205 \pm 0.005$ |
| 0.02 | SOCOP (ours) | $0.950 \pm 0.002$ | $2.326 \pm 0.060$ | $0.212 \pm 0.005$ |
| 0.03 | SOCOP (ours) | $0.950 \pm 0.002$ | $2.122 \pm 0.053$ | $0.217 \pm 0.006$ |
| 0.04 | SOCOP (ours) | $0.950 \pm 0.002$ | $2.012 \pm 0.042$ | $0.221 \pm 0.005$ |
| 0.05 | SOCOP (ours) | $0.950 \pm 0.002$ | $1.946 \pm 0.038$ | $0.226 \pm 0.005$ |
| 0.06 | SOCOP (ours) | $0.950 \pm 0.002$ | $1.898 \pm 0.035$ | $0.229 \pm 0.005$ |
| 0.07 | SOCOP (ours) | $0.950 \pm 0.002$ | $1.859 \pm 0.034$ | $0.232 \pm 0.006$ |
| 0.08 | SOCOP (ours) | $0.950 \pm 0.002$ | $1.828 \pm 0.033$ | $0.234 \pm 0.006$ |
| 0.09 | SOCOP (ours) | $0.950 \pm 0.002$ | $1.802 \pm 0.031$ | $0.236 \pm 0.006$ |
| 0.10 | SOCOP (ours) | $0.950 \pm 0.002$ | $1.782 \pm 0.028$ | $0.238 \pm 0.005$ |
| 0.20 | SOCOP (ours) | $0.950 \pm 0.002$ | $1.678 \pm 0.023$ | $0.255 \pm 0.005$ |
| 0.30 | SOCOP (ours) | $0.950 \pm 0.002$ | $1.640 \pm 0.022$ | $0.265 \pm 0.006$ |
| 0.40 | SOCOP (ours) | $0.950 \pm 0.002$ | $1.621 \pm 0.021$ | $0.274 \pm 0.006$ |
| 0.50 | SOCOP (ours) | $0.950 \pm 0.002$ | $1.608 \pm 0.023$ | $0.279 \pm 0.006$ |
| 0.60 | SOCOP (ours) | $0.950 \pm 0.002$ | $1.596 \pm 0.023$ | $0.283 \pm 0.007$ |
| 0.70 | SOCOP (ours) | $0.950 \pm 0.002$ | $1.587 \pm 0.022$ | $0.286 \pm 0.007$ |
| 0.80 | SOCOP (ours) | $0.950 \pm 0.002$ | $1.581 \pm 0.022$ | $0.289 \pm 0.006$ |
| 0.90 | SOCOP (ours) | $0.950 \pm 0.002$ | $1.575 \pm 0.022$ | $0.291 \pm 0.007$ |
| 1.00 | SOCOP (ours) | $0.950 \pm 0.002$ | $1.571 \pm 0.022$ | $0.293 \pm 0.007$ |
| $\infty$ | Least Ambiguous Sets | $0.950 \pm 0.002$ | $1.542 \pm 0.018$ | $0.329 \pm 0.007$ |

Table 8: Performance of `ConvNeXt-base` on ImageNet-Val with different $\lambda$ values ($\alpha = 0.05$). Results are averaged over 100 data splits.

| $\lambda$ | Method | Coverage | Avg Size | $P(\text{size} > 1)$ |
|---|---|---|---|---|
| 0 | Pure Singleton | $0.950 \pm 0.002$ | $226.991 \pm 4.935$ | $0.226 \pm 0.005$ |
| 0.01 | SOCOP (ours) | $0.950 \pm 0.002$ | $4.074 \pm 0.155$ | $0.244 \pm 0.005$ |
| 0.02 | SOCOP (ours) | $0.950 \pm 0.002$ | $3.194 \pm 0.097$ | $0.254 \pm 0.005$ |
| 0.03 | SOCOP (ours) | $0.950 \pm 0.002$ | $2.866 \pm 0.081$ | $0.262 \pm 0.005$ |
| 0.04 | SOCOP (ours) | $0.950 \pm 0.002$ | $2.683 \pm 0.068$ | $0.268 \pm 0.005$ |
| 0.05 | SOCOP (ours) | $0.950 \pm 0.002$ | $2.567 \pm 0.067$ | $0.274 \pm 0.005$ |
| 0.06 | SOCOP (ours) | $0.950 \pm 0.002$ | $2.485 \pm 0.065$ | $0.278 \pm 0.006$ |
| 0.07 | SOCOP (ours) | $0.950 \pm 0.002$ | $2.429 \pm 0.062$ | $0.283 \pm 0.006$ |
| 0.08 | SOCOP (ours) | $0.950 \pm 0.002$ | $2.383 \pm 0.063$ | $0.287 \pm 0.006$ |
| 0.09 | SOCOP (ours) | $0.950 \pm 0.002$ | $2.344 \pm 0.060$ | $0.290 \pm 0.006$ |
| 0.10 | SOCOP (ours) | $0.949 \pm 0.002$ | $2.307 \pm 0.056$ | $0.293 \pm 0.006$ |
| 0.20 | SOCOP (ours) | $0.950 \pm 0.002$ | $2.121 \pm 0.045$ | $0.310 \pm 0.006$ |
| 0.30 | SOCOP (ours) | $0.950 \pm 0.002$ | $2.044 \pm 0.036$ | $0.321 \pm 0.006$ |
| 0.40 | SOCOP (ours) | $0.950 \pm 0.002$ | $2.008 \pm 0.036$ | $0.330 \pm 0.006$ |
| 0.50 | SOCOP (ours) | $0.950 \pm 0.002$ | $1.983 \pm 0.035$ | $0.336 \pm 0.006$ |
| 0.60 | SOCOP (ours) | $0.950 \pm 0.002$ | $1.964 \pm 0.033$ | $0.341 \pm 0.005$ |
| 0.70 | SOCOP (ours) | $0.950 \pm 0.002$ | $1.951 \pm 0.035$ | $0.345 \pm 0.006$ |
| 0.80 | SOCOP (ours) | $0.950 \pm 0.002$ | $1.943 \pm 0.035$ | $0.349 \pm 0.006$ |
| 0.90 | SOCOP (ours) | $0.950 \pm 0.002$ | $1.937 \pm 0.034$ | $0.352 \pm 0.006$ |
| 1.00 | SOCOP (ours) | $0.950 \pm 0.002$ | $1.932 \pm 0.033$ | $0.355 \pm 0.006$ |
| $\infty$ | Least Ambiguous Sets | $0.950 \pm 0.002$ | $1.897 \pm 0.034$ | $0.398 \pm 0.007$ |

Table 9: Performance of `Swin-v2-b` on ImageNet-Val with different $\lambda$ values ($\alpha = 0.05$). Results are averaged over 100 data splits.

| $\lambda$ | Method | Coverage | Avg Size | $P(\text{size} > 1)$ |
|---|---|---|---|---|
| 0 | Pure Singleton | $0.950 \pm 0.002$ | $225.685 \pm 4.909$ | $0.225 \pm 0.005$ |
| 0.01 | SOCOP (ours) | $0.949 \pm 0.002$ | $3.844 \pm 0.121$ | $0.245 \pm 0.004$ |
| 0.02 | SOCOP (ours) | $0.949 \pm 0.002$ | $3.089 \pm 0.087$ | $0.256 \pm 0.005$ |
| 0.03 | SOCOP (ours) | $0.949 \pm 0.002$ | $2.775 \pm 0.070$ | $0.262 \pm 0.005$ |
| 0.04 | SOCOP (ours) | $0.949 \pm 0.002$ | $2.610 \pm 0.060$ | $0.268 \pm 0.005$ |
| 0.05 | SOCOP (ours) | $0.949 \pm 0.002$ | $2.502 \pm 0.054$ | $0.273 \pm 0.005$ |
| 0.06 | SOCOP (ours) | $0.949 \pm 0.002$ | $2.426 \pm 0.052$ | $0.278 \pm 0.005$ |
| 0.07 | SOCOP (ours) | $0.949 \pm 0.002$ | $2.369 \pm 0.056$ | $0.281 \pm 0.005$ |
| 0.08 | SOCOP (ours) | $0.949 \pm 0.002$ | $2.324 \pm 0.053$ | $0.284 \pm 0.005$ |
| 0.09 | SOCOP (ours) | $0.949 \pm 0.002$ | $2.285 \pm 0.051$ | $0.287 \pm 0.005$ |
| 0.10 | SOCOP (ours) | $0.949 \pm 0.002$ | $2.253 \pm 0.051$ | $0.289 \pm 0.006$ |
| 0.20 | SOCOP (ours) | $0.949 \pm 0.002$ | $2.096 \pm 0.045$ | $0.309 \pm 0.006$ |
| 0.30 | SOCOP (ours) | $0.949 \pm 0.002$ | $2.030 \pm 0.038$ | $0.320 \pm 0.006$ |
| 0.40 | SOCOP (ours) | $0.949 \pm 0.002$ | $1.997 \pm 0.036$ | $0.329 \pm 0.006$ |
| 0.50 | SOCOP (ours) | $0.949 \pm 0.002$ | $1.976 \pm 0.035$ | $0.336 \pm 0.006$ |
| 0.60 | SOCOP (ours) | $0.950 \pm 0.002$ | $1.961 \pm 0.035$ | $0.341 \pm 0.006$ |
| 0.70 | SOCOP (ours) | $0.950 \pm 0.002$ | $1.949 \pm 0.036$ | $0.345 \pm 0.006$ |
| 0.80 | SOCOP (ours) | $0.950 \pm 0.002$ | $1.942 \pm 0.037$ | $0.349 \pm 0.007$ |
| 0.90 | SOCOP (ours) | $0.950 \pm 0.002$ | $1.935 \pm 0.037$ | $0.352 \pm 0.007$ |
| 1.00 | SOCOP (ours) | $0.950 \pm 0.002$ | $1.931 \pm 0.037$ | $0.355 \pm 0.007$ |
| $\infty$ | Least Ambiguous Sets | $0.950 \pm 0.002$ | $1.881 \pm 0.033$ | $0.396 \pm 0.007$ |

Table 10: Performance of `ViT-h-14` on ImageNet-Val with different $\lambda$ values ($\alpha = 0.05$). Results are averaged over 100 data splits.

| $\lambda$ | Method | Coverage | Avg Size | $P(\text{size} > 1)$ |
|---|---|---|---|---|
| 0 | Pure Singleton | $0.950 \pm 0.002$ | $136.219 \pm 4.980$ | $0.135 \pm 0.005$ |
| 0.01 | SOCOP (ours) | $0.950 \pm 0.003$ | $2.061 \pm 0.062$ | $0.141 \pm 0.005$ |
| 0.02 | SOCOP (ours) | $0.950 \pm 0.003$ | $1.761 \pm 0.040$ | $0.145 \pm 0.005$ |
| 0.03 | SOCOP (ours) | $0.950 \pm 0.002$ | $1.641 \pm 0.030$ | $0.148 \pm 0.005$ |
| 0.04 | SOCOP (ours) | $0.950 \pm 0.002$ | $1.572 \pm 0.025$ | $0.151 \pm 0.005$ |
| 0.05 | SOCOP (ours) | $0.950 \pm 0.002$ | $1.529 \pm 0.023$ | $0.153 \pm 0.005$ |
| 0.06 | SOCOP (ours) | $0.950 \pm 0.002$ | $1.498 \pm 0.023$ | $0.155 \pm 0.005$ |
| 0.07 | SOCOP (ours) | $0.950 \pm 0.002$ | $1.474 \pm 0.022$ | $0.156 \pm 0.005$ |
| 0.08 | SOCOP (ours) | $0.950 \pm 0.002$ | $1.457 \pm 0.021$ | $0.158 \pm 0.005$ |
| 0.09 | SOCOP (ours) | $0.950 \pm 0.002$ | $1.444 \pm 0.021$ | $0.159 \pm 0.005$ |
| 0.10 | SOCOP (ours) | $0.950 \pm 0.002$ | $1.432 \pm 0.022$ | $0.161 \pm 0.005$ |
| 0.20 | SOCOP (ours) | $0.950 \pm 0.002$ | $1.368 \pm 0.016$ | $0.171 \pm 0.005$ |
| 0.30 | SOCOP (ours) | $0.950 \pm 0.002$ | $1.344 \pm 0.015$ | $0.177 \pm 0.005$ |
| 0.40 | SOCOP (ours) | $0.950 \pm 0.002$ | $1.332 \pm 0.014$ | $0.183 \pm 0.005$ |
| 0.50 | SOCOP (ours) | $0.950 \pm 0.002$ | $1.324 \pm 0.014$ | $0.187 \pm 0.005$ |
| 0.60 | SOCOP (ours) | $0.950 \pm 0.002$ | $1.319 \pm 0.013$ | $0.190 \pm 0.005$ |
| 0.70 | SOCOP (ours) | $0.950 \pm 0.002$ | $1.315 \pm 0.012$ | $0.193 \pm 0.005$ |
| 0.80 | SOCOP (ours) | $0.950 \pm 0.002$ | $1.312 \pm 0.013$ | $0.195 \pm 0.006$ |
| 0.90 | SOCOP (ours) | $0.950 \pm 0.002$ | $1.309 \pm 0.014$ | $0.197 \pm 0.006$ |
| 1.00 | SOCOP (ours) | $0.950 \pm 0.002$ | $1.307 \pm 0.014$ | $0.198 \pm 0.006$ |
| $\infty$ | Least Ambiguous Sets | $0.950 \pm 0.002$ | $1.291 \pm 0.011$ | $0.224 \pm 0.006$ |

Table 11: Performance on ImageNet-V2, with a protocol identical to that in Table 1

| Model | Method | Coverage | Avg Size | $P(\text{size} > 1)$ |
|---|---|---|---|---|
| ResNet152-v2 | Plug-In | $0.975 \pm 0.004$ | $190.453 \pm 23.837$ | $0.839 \pm 0.035$ |
| | RAPS | $0.950 \pm 0.004$ | $11.524 \pm 0.793$ | $1.000 \pm 0.000$ |
| | Pure Singleton | $0.950 \pm 0.005$ | $432.673 \pm 12.869$ | $0.432 \pm 0.013$ |
| | Least Ambiguous Sets | $0.949 \pm 0.005$ | $9.067 \pm 0.677$ | $0.798 \pm 0.011$ |
| | **SOCOP** (ours) | $0.950 \pm 0.005$ | $10.212 \pm 1.099$ | $0.655 \pm 0.031$ |
| EfficientNet-v2-l | Plug-In | $0.963 \pm 0.004$ | $70.999 \pm 7.761$ | $0.661 \pm 0.022$ |
| | RAPS | $0.950 \pm 0.005$ | $5.947 \pm 0.485$ | $0.920 \pm 0.069$ |
| | Pure Singleton | $0.950 \pm 0.005$ | $369.726 \pm 14.194$ | $0.369 \pm 0.014$ |
| | Least Ambiguous Sets | $0.950 \pm 0.004$ | $4.157 \pm 0.231$ | $0.718 \pm 0.016$ |
| | **SOCOP** (ours) | $0.950 \pm 0.004$ | $4.736 \pm 0.354$ | $0.571 \pm 0.024$ |
| ConvNeXt-base | Plug-In | $0.971 \pm 0.005$ | $161.625 \pm 22.907$ | $0.845 \pm 0.030$ |
| | RAPS | $0.950 \pm 0.005$ | $10.380 \pm 0.819$ | $1.000 \pm 0.000$ |
| | Pure Singleton | $0.950 \pm 0.004$ | $428.852 \pm 14.761$ | $0.428 \pm 0.015$ |
| | Least Ambiguous Sets | $0.950 \pm 0.005$ | $6.810 \pm 0.492$ | $0.787 \pm 0.016$ |
| | **SOCOP** (ours) | $0.950 \pm 0.005$ | $7.578 \pm 0.558$ | $0.629 \pm 0.016$ |
| Swin-v2-b | Plug-In | $0.981 \pm 0.004$ | $166.462 \pm 26.307$ | $0.941 \pm 0.032$ |
| | RAPS | $0.951 \pm 0.005$ | $9.306 \pm 0.860$ | $1.000 \pm 0.000$ |
| | Pure Singleton | $0.950 \pm 0.005$ | $414.604 \pm 13.283$ | $0.414 \pm 0.013$ |
| | Least Ambiguous Sets | $0.950 \pm 0.004$ | $6.673 \pm 0.472$ | $0.777 \pm 0.017$ |
| | **SOCOP** (ours) | $0.950 \pm 0.005$ | $7.634 \pm 0.703$ | $0.626 \pm 0.021$ |
| ViT-h-14 | Plug-In | $0.965 \pm 0.003$ | $33.017 \pm 3.027$ | $0.540 \pm 0.013$ |
| | RAPS | $0.951 \pm 0.005$ | $3.259 \pm 0.264$ | $0.979 \pm 0.092$ |
| | Pure Singleton | $0.950 \pm 0.004$ | $304.159 \pm 13.851$ | $0.304 \pm 0.014$ |
| | Least Ambiguous Sets | $0.950 \pm 0.005$ | $2.378 \pm 0.105$ | $0.539 \pm 0.018$ |
| | **SOCOP** (ours) | $0.950 \pm 0.005$ | $2.695 \pm 0.165$ | $0.421 \pm 0.024$ |

Table 12: Performance of ResNet152-v2 on ImageNet-V2 with different $\lambda$ values ($\alpha = 0.05$). Results are averaged over 100 data splits.

| $\lambda$ | Method | Coverage | Avg Size | $P(\text{size} > 1)$ |
|---|---|---|---|---|
| 0 | Pure Singleton | $0.950 \pm 0.005$ | $432.673 \pm 12.869$ | $0.432 \pm 0.013$ |
| 0.01 | SOCOP (ours) | $0.950 \pm 0.006$ | $25.474 \pm 3.292$ | $0.502 \pm 0.013$ |
| 0.02 | SOCOP (ours) | $0.950 \pm 0.005$ | $16.743 \pm 1.657$ | $0.531 \pm 0.013$ |
| 0.03 | SOCOP (ours) | $0.950 \pm 0.005$ | $14.088 \pm 1.270$ | $0.550 \pm 0.012$ |
| 0.04 | SOCOP (ours) | $0.950 \pm 0.005$ | $13.112 \pm 1.138$ | $0.567 \pm 0.012$ |
| 0.05 | SOCOP (ours) | $0.950 \pm 0.005$ | $12.527 \pm 1.047$ | $0.581 \pm 0.013$ |
| 0.06 | SOCOP (ours) | $0.950 \pm 0.005$ | $12.091 \pm 1.020$ | $0.592 \pm 0.013$ |
| 0.07 | SOCOP (ours) | $0.950 \pm 0.005$ | $11.732 \pm 0.974$ | $0.601 \pm 0.013$ |
| 0.08 | SOCOP (ours) | $0.950 \pm 0.005$ | $11.432 \pm 0.931$ | $0.609 \pm 0.013$ |
| 0.09 | SOCOP (ours) | $0.950 \pm 0.005$ | $11.171 \pm 0.876$ | $0.615 \pm 0.012$ |
| 0.10 | SOCOP (ours) | $0.950 \pm 0.005$ | $10.920 \pm 0.826$ | $0.620 \pm 0.012$ |
| 0.20 | SOCOP (ours) | $0.950 \pm 0.005$ | $9.949 \pm 0.685$ | $0.660 \pm 0.011$ |
| 0.30 | SOCOP (ours) | $0.950 \pm 0.005$ | $9.745 \pm 0.661$ | $0.684 \pm 0.011$ |
| 0.40 | SOCOP (ours) | $0.950 \pm 0.005$ | $9.616 \pm 0.623$ | $0.699 \pm 0.010$ |
| 0.50 | SOCOP (ours) | $0.950 \pm 0.005$ | $9.555 \pm 0.609$ | $0.711 \pm 0.010$ |
| 0.60 | SOCOP (ours) | $0.950 \pm 0.005$ | $9.508 \pm 0.579$ | $0.720 \pm 0.010$ |
| 0.70 | SOCOP (ours) | $0.950 \pm 0.005$ | $9.427 \pm 0.543$ | $0.726 \pm 0.010$ |
| 0.80 | SOCOP (ours) | $0.950 \pm 0.005$ | $9.344 \pm 0.554$ | $0.730 \pm 0.010$ |
| 0.90 | SOCOP (ours) | $0.950 \pm 0.005$ | $9.290 \pm 0.564$ | $0.735 \pm 0.010$ |
| 1.00 | SOCOP (ours) | $0.950 \pm 0.005$ | $9.262 \pm 0.578$ | $0.739 \pm 0.011$ |
| $\infty$ | Least Ambiguous Sets | $0.949 \pm 0.005$ | $9.067 \pm 0.677$ | $0.798 \pm 0.011$ |

Table 13: Performance of `EfficientNet-v2-l` on ImageNet-V2 with different $\lambda$ values ($\alpha = 0.05$). Results are averaged over 100 data splits.

| $\lambda$ | Method | Coverage | Avg Size | $P(\text{size} > 1)$ |
|---|---|---|---|---|
| 0 | Pure Singleton | $0.950 \pm 0.005$ | $369.726 \pm 14.194$ | $0.369 \pm 0.014$ |
| 0.01 | SOCOP (ours) | $0.951 \pm 0.005$ | $12.243 \pm 1.247$ | $0.422 \pm 0.014$ |
| 0.02 | SOCOP (ours) | $0.950 \pm 0.004$ | $8.506 \pm 0.744$ | $0.448 \pm 0.015$ |
| 0.03 | SOCOP (ours) | $0.950 \pm 0.004$ | $7.266 \pm 0.505$ | $0.466 \pm 0.013$ |
| 0.04 | SOCOP (ours) | $0.950 \pm 0.004$ | $6.643 \pm 0.433$ | $0.480 \pm 0.013$ |
| 0.05 | SOCOP (ours) | $0.950 \pm 0.004$ | $6.216 \pm 0.413$ | $0.491 \pm 0.014$ |
| 0.06 | SOCOP (ours) | $0.950 \pm 0.004$ | $5.910 \pm 0.398$ | $0.500 \pm 0.015$ |
| 0.07 | SOCOP (ours) | $0.950 \pm 0.004$ | $5.718 \pm 0.397$ | $0.509 \pm 0.016$ |
| 0.08 | SOCOP (ours) | $0.950 \pm 0.004$ | $5.559 \pm 0.399$ | $0.516 \pm 0.016$ |
| 0.09 | SOCOP (ours) | $0.950 \pm 0.004$ | $5.429 \pm 0.405$ | $0.522 \pm 0.017$ |
| 0.10 | SOCOP (ours) | $0.950 \pm 0.004$ | $5.304 \pm 0.385$ | $0.528 \pm 0.016$ |
| 0.20 | SOCOP (ours) | $0.950 \pm 0.004$ | $4.751 \pm 0.302$ | $0.565 \pm 0.016$ |
| 0.30 | SOCOP (ours) | $0.950 \pm 0.004$ | $4.586 \pm 0.285$ | $0.590 \pm 0.016$ |
| 0.40 | SOCOP (ours) | $0.950 \pm 0.004$ | $4.496 \pm 0.290$ | $0.606 \pm 0.016$ |
| 0.50 | SOCOP (ours) | $0.950 \pm 0.004$ | $4.418 \pm 0.273$ | $0.617 \pm 0.016$ |
| 0.60 | SOCOP (ours) | $0.950 \pm 0.004$ | $4.372 \pm 0.258$ | $0.626 \pm 0.015$ |
| 0.70 | SOCOP (ours) | $0.950 \pm 0.004$ | $4.335 \pm 0.251$ | $0.634 \pm 0.015$ |
| 0.80 | SOCOP (ours) | $0.950 \pm 0.004$ | $4.305 \pm 0.248$ | $0.639 \pm 0.015$ |
| 0.90 | SOCOP (ours) | $0.950 \pm 0.004$ | $4.284 \pm 0.250$ | $0.644 \pm 0.015$ |
| 1.00 | SOCOP (ours) | $0.950 \pm 0.004$ | $4.272 \pm 0.249$ | $0.649 \pm 0.016$ |
| $\infty$ | Least Ambiguous Sets | $0.950 \pm 0.004$ | $4.157 \pm 0.231$ | $0.718 \pm 0.016$ |

Table 14: Performance of `ConvNeXt-base` on ImageNet-V2 with different $\lambda$ values ($\alpha = 0.05$). Results are averaged over 100 data splits.

| $\lambda$ | Method | Coverage | Avg Size | $P(\text{size} > 1)$ |
|---|---|---|---|---|
| 0 | Pure Singleton | $0.950 \pm 0.004$ | $428.852 \pm 14.761$ | $0.428 \pm 0.015$ |
| 0.01 | SOCOP (ours) | $0.950 \pm 0.004$ | $19.963 \pm 2.012$ | $0.465 \pm 0.012$ |
| 0.02 | SOCOP (ours) | $0.950 \pm 0.004$ | $14.093 \pm 1.081$ | $0.498 \pm 0.011$ |
| 0.03 | SOCOP (ours) | $0.950 \pm 0.004$ | $11.929 \pm 0.907$ | $0.518 \pm 0.012$ |
| 0.04 | SOCOP (ours) | $0.950 \pm 0.004$ | $10.687 \pm 0.818$ | $0.531 \pm 0.012$ |
| 0.05 | SOCOP (ours) | $0.950 \pm 0.004$ | $9.976 \pm 0.750$ | $0.543 \pm 0.012$ |
| 0.06 | SOCOP (ours) | $0.950 \pm 0.004$ | $9.448 \pm 0.682$ | $0.552 \pm 0.012$ |
| 0.07 | SOCOP (ours) | $0.950 \pm 0.004$ | $9.104 \pm 0.634$ | $0.560 \pm 0.012$ |
| 0.08 | SOCOP (ours) | $0.950 \pm 0.004$ | $8.830 \pm 0.599$ | $0.568 \pm 0.011$ |
| 0.09 | SOCOP (ours) | $0.950 \pm 0.004$ | $8.649 \pm 0.577$ | $0.574 \pm 0.011$ |
| 0.10 | SOCOP (ours) | $0.950 \pm 0.004$ | $8.493 \pm 0.561$ | $0.580 \pm 0.012$ |
| 0.20 | SOCOP (ours) | $0.950 \pm 0.005$ | $7.668 \pm 0.525$ | $0.622 \pm 0.013$ |
| 0.30 | SOCOP (ours) | $0.950 \pm 0.005$ | $7.386 \pm 0.494$ | $0.647 \pm 0.013$ |
| 0.40 | SOCOP (ours) | $0.950 \pm 0.005$ | $7.245 \pm 0.505$ | $0.665 \pm 0.014$ |
| 0.50 | SOCOP (ours) | $0.950 \pm 0.005$ | $7.182 \pm 0.503$ | $0.677 \pm 0.014$ |
| 0.60 | SOCOP (ours) | $0.950 \pm 0.005$ | $7.148 \pm 0.513$ | $0.688 \pm 0.015$ |
| 0.70 | SOCOP (ours) | $0.950 \pm 0.005$ | $7.135 \pm 0.510$ | $0.697 \pm 0.015$ |
| 0.80 | SOCOP (ours) | $0.950 \pm 0.005$ | $7.111 \pm 0.519$ | $0.704 \pm 0.016$ |
| 0.90 | SOCOP (ours) | $0.950 \pm 0.005$ | $7.108 \pm 0.512$ | $0.711 \pm 0.016$ |
| 1.00 | SOCOP (ours) | $0.950 \pm 0.005$ | $7.104 \pm 0.509$ | $0.717 \pm 0.016$ |
| $\infty$ | Least Ambiguous Sets | $0.950 \pm 0.005$ | $6.810 \pm 0.492$ | $0.787 \pm 0.016$ |

Table 15: Performance of `Swin-v2-b` on ImageNet-V2 with different $\lambda$ values ($\alpha = 0.05$). Results are averaged over 100 data splits.

| $\lambda$ | Method | Coverage | Avg Size | $P(\text{size} > 1)$ |
|---|---|---|---|---|
| 0 | Pure Singleton | $0.950 \pm 0.005$ | $414.604 \pm 13.283$ | $0.414 \pm 0.013$ |
| 0.01 | SOCOP (ours) | $0.951 \pm 0.004$ | $20.127 \pm 1.710$ | $0.478 \pm 0.011$ |
| 0.02 | SOCOP (ours) | $0.950 \pm 0.005$ | $13.318 \pm 1.226$ | $0.501 \pm 0.013$ |
| 0.03 | SOCOP (ours) | $0.950 \pm 0.005$ | $11.647 \pm 1.024$ | $0.522 \pm 0.014$ |
| 0.04 | SOCOP (ours) | $0.950 \pm 0.004$ | $10.768 \pm 0.875$ | $0.537 \pm 0.013$ |
| 0.05 | SOCOP (ours) | $0.950 \pm 0.004$ | $10.036 \pm 0.794$ | $0.547 \pm 0.013$ |
| 0.06 | SOCOP (ours) | $0.950 \pm 0.004$ | $9.507 \pm 0.706$ | $0.556 \pm 0.013$ |
| 0.07 | SOCOP (ours) | $0.950 \pm 0.004$ | $9.113 \pm 0.620$ | $0.564 \pm 0.013$ |
| 0.08 | SOCOP (ours) | $0.950 \pm 0.004$ | $8.834 \pm 0.548$ | $0.571 \pm 0.012$ |
| 0.09 | SOCOP (ours) | $0.950 \pm 0.004$ | $8.640 \pm 0.548$ | $0.579 \pm 0.013$ |
| 0.10 | SOCOP (ours) | $0.950 \pm 0.004$ | $8.450 \pm 0.529$ | $0.585 \pm 0.013$ |
| 0.20 | SOCOP (ours) | $0.950 \pm 0.004$ | $7.625 \pm 0.596$ | $0.624 \pm 0.015$ |
| 0.30 | SOCOP (ours) | $0.950 \pm 0.005$ | $7.337 \pm 0.583$ | $0.646 \pm 0.016$ |
| 0.40 | SOCOP (ours) | $0.950 \pm 0.004$ | $7.183 \pm 0.570$ | $0.662 \pm 0.016$ |
| 0.50 | SOCOP (ours) | $0.950 \pm 0.005$ | $7.093 \pm 0.563$ | $0.674 \pm 0.017$ |
| 0.60 | SOCOP (ours) | $0.950 \pm 0.005$ | $7.049 \pm 0.568$ | $0.684 \pm 0.017$ |
| 0.70 | SOCOP (ours) | $0.950 \pm 0.005$ | $7.004 \pm 0.551$ | $0.692 \pm 0.017$ |
| 0.80 | SOCOP (ours) | $0.950 \pm 0.005$ | $6.971 \pm 0.549$ | $0.699 \pm 0.017$ |
| 0.90 | SOCOP (ours) | $0.950 \pm 0.005$ | $6.954 \pm 0.538$ | $0.705 \pm 0.017$ |
| 1.00 | SOCOP (ours) | $0.950 \pm 0.005$ | $6.929 \pm 0.527$ | $0.710 \pm 0.017$ |
| $\infty$ | Least Ambiguous Sets | $0.950 \pm 0.004$ | $6.673 \pm 0.472$ | $0.777 \pm 0.017$ |

Table 16: Performance of `ViT-h-14` on ImageNet-V2 with different $\lambda$ values ($\alpha = 0.05$). Results are averaged over 100 data splits.

| $\lambda$ | Method | Coverage | Avg Size | $P(\text{size} > 1)$ |
|---|---|---|---|---|
| 0.00 | Pure Singleton | $0.950 \pm 0.004$ | $304.159 \pm 13.851$ | $0.304 \pm 0.014$ |
| 0.01 | SOCOP (ours) | $0.950 \pm 0.004$ | $6.672 \pm 0.461$ | $0.323 \pm 0.012$ |
| 0.02 | SOCOP (ours) | $0.950 \pm 0.004$ | $4.803 \pm 0.258$ | $0.336 \pm 0.011$ |
| 0.03 | SOCOP (ours) | $0.950 \pm 0.004$ | $4.145 \pm 0.213$ | $0.346 \pm 0.011$ |
| 0.04 | SOCOP (ours) | $0.950 \pm 0.004$ | $3.747 \pm 0.200$ | $0.352 \pm 0.012$ |
| 0.05 | SOCOP (ours) | $0.950 \pm 0.004$ | $3.493 \pm 0.186$ | $0.357 \pm 0.012$ |
| 0.06 | SOCOP (ours) | $0.950 \pm 0.004$ | $3.329 \pm 0.176$ | $0.361 \pm 0.013$ |
| 0.07 | SOCOP (ours) | $0.950 \pm 0.004$ | $3.223 \pm 0.162$ | $0.367 \pm 0.013$ |
| 0.08 | SOCOP (ours) | $0.950 \pm 0.005$ | $3.147 \pm 0.160$ | $0.372 \pm 0.013$ |
| 0.09 | SOCOP (ours) | $0.950 \pm 0.005$ | $3.084 \pm 0.165$ | $0.377 \pm 0.014$ |
| 0.10 | SOCOP (ours) | $0.950 \pm 0.005$ | $3.027 \pm 0.167$ | $0.381 \pm 0.015$ |
| 0.20 | SOCOP (ours) | $0.950 \pm 0.005$ | $2.743 \pm 0.142$ | $0.411 \pm 0.016$ |
| 0.30 | SOCOP (ours) | $0.950 \pm 0.005$ | $2.636 \pm 0.141$ | $0.429 \pm 0.016$ |
| 0.40 | SOCOP (ours) | $0.950 \pm 0.005$ | $2.575 \pm 0.141$ | $0.441 \pm 0.017$ |
| 0.50 | SOCOP (ours) | $0.950 \pm 0.005$ | $2.539 \pm 0.135$ | $0.450 \pm 0.017$ |
| 0.60 | SOCOP (ours) | $0.951 \pm 0.005$ | $2.516 \pm 0.127$ | $0.458 \pm 0.017$ |
| 0.70 | SOCOP (ours) | $0.950 \pm 0.005$ | $2.496 \pm 0.123$ | $0.464 \pm 0.017$ |
| 0.80 | SOCOP (ours) | $0.950 \pm 0.005$ | $2.480 \pm 0.117$ | $0.469 \pm 0.016$ |
| 0.90 | SOCOP (ours) | $0.950 \pm 0.005$ | $2.471 \pm 0.116$ | $0.474 \pm 0.016$ |
| 1.00 | SOCOP (ours) | $0.950 \pm 0.005$ | $2.461 \pm 0.116$ | $0.478 \pm 0.016$ |
| $\infty$ | Least Ambiguous Sets | $0.950 \pm 0.005$ | $2.378 \pm 0.105$ | $0.539 \pm 0.018$ |

### D.3 CPL Method

We report the performance of the `CPL` method (Kiyani et al., 2024) under the same experimental protocol as in the main text. Following Kiyani et al. (2024), we implement $\mathcal{H}$ as a linear head on top of the pre-trained model, mapping the final hidden-layer representations to a real-valued scalar. The results, shown in Table 17, indicate that this method exhibits slight undercoverage, while attaining similar performance to `Least Ambiguous Sets`. Notably, these results are different from the ones reported by (Kiyani et al., 2024), where the `CPL` method reduced average set sizes. However, the experimental settings considered in the two papers are different, which may explain the experimental differences. In particular, their results use older large language models which perform quite poorly on MMLU, such that the original average set sizes are very large, being for instance equal to approximately 3.5 out of 4 in one example. This leaves ample opportunity for improving the set sizes by the `CPL` method. In contrast, in our setting, the language models have a higher performance (leading to smaller set sizes with the default least ambiguous set sizes method, around 2.5 out of 4), which may leave less opportunity for improvement.

### E Hyperparameter Grid

For `RAPS`, we follow Angelopoulos et al. (2021), using the gird $\lambda \in \{0.001, 0.01, 0.1, 0.2, 0.5\}$ to optimize set size and a grid with smaller values $\lambda \in \{0.00001, 0.0001, 0.0008, 0.001, 0.0015, 0.002\}$ to optimize SSCV. For our `SOCOP` method, we use a linearly spaced grid of 15 values over over $[0.05, 1.0]$ to optimize the balance between set size and non-singleton rate, and a linearly spaced grid of 15 values over $[0.005, 0.1]$ to optimize SSCV.

Table 17: Performance of `CPL` (Kiyani et al., 2024) on ImageNet-Val, ImageNet-V2, TissueMNIST and MMLU with the same protocol used ($\alpha = 0.05$).

| Model | Coverage | Avg Size | $P(\text{size} > 1)$ |
|---|---|---|---|
| ResNet152-v2 | $0.950 \pm 0.002$ | $2.297 \pm 0.059$ | $0.463 \pm 0.006$ |
| EfficientNet-v2-l | $0.949 \pm 0.003$ | $1.542 \pm 0.044$ | $0.327 \pm 0.011$ |
| ConvNeXt-base | $0.948 \pm 0.003$ | $1.866 \pm 0.050$ | $0.392 \pm 0.011$ |
| Swin-v2-b | $0.948 \pm 0.003$ | $1.841 \pm 0.039$ | $0.386 \pm 0.009$ |
| ViT-h-14 | $0.949 \pm 0.004$ | $1.292 \pm 0.030$ | $0.221 \pm 0.017$ |

(a) ImageNet-Val

| Model | Coverage | Avg Size | $P(\text{size} > 1)$ |
|---|---|---|---|
| ResNet152-v2 | $0.950 \pm 0.006$ | $9.295 \pm 1.167$ | $0.797 \pm 0.016$ |
| EfficientNet-v2-l | $0.940 \pm 0.006$ | $3.394 \pm 0.400$ | $0.675 \pm 0.017$ |
| ConvNeXt-base | $0.949 \pm 0.005$ | $6.677 \pm 0.615$ | $0.773 \pm 0.019$ |
| Swin-v2-b | $0.949 \pm 0.005$ | $6.493 \pm 0.673$ | $0.764 \pm 0.024$ |
| ViT-h-14 | $0.948 \pm 0.005$ | $2.400 \pm 0.154$ | $0.489 \pm 0.019$ |

(b) ImageNet-V2

| Model | Coverage | Avg Size | $P(\text{size} > 1)$ |
|---|---|---|---|
| ResNet-50 (224) | $0.950 \pm 0.003$ | $2.640 \pm 0.040$ | $0.791 \pm 0.008$ |

(c) TissueMNIST

| Model | Coverage | Avg Size | $P(\text{size} > 1)$ |
|---|---|---|---|
| Llama3.1-8B-Instruct | $0.948 \pm 0.006$ | $2.400 \pm 0.046$ | $0.644 \pm 0.012$ |

(d) MMLU

