# OpenReview forum: "Singleton-Optimized Conformal Prediction"
_ICLR.cc/2026/Conference — ICLR 2026 Poster_

### Official Review · Reviewer_67Nw · 2025-10-21

**Soundness:** 3
**Presentation:** 3
**Contribution:** 2
**Rating:** 6
**Confidence:** 4

**Summary:**

**Summary**

This paper addresses a key limitation of conformal prediction: while it creates reliable prediction sets (e.g., a list of possible answers that is 95% likely to contain the true one), these sets often contain multiple items, making them ambiguous for automated decision-making. The authors argue that the most useful outcome is a set with just one answer (a "singleton"), as it provides a clear, actionable result.
They introduce a method called SOCOP (Singleton-Optimized Conformal Prediction) which is specifically designed to maximize the frequency of these singleton predictions.
Instead of only trying to make the prediction sets as small as possible on average, SOCOP try to produce unambiguous, single-answer outputs, creating a more practical trade-off for real-world applications where decision clarity is more important than a slightly smaller average set size.


**Contribution**

The main contribution is a novel and highly efficient method to achieve this goal.
The authors derive a new "nonconformity score" from an optimization objective that directly penalizes non-singleton sets.
Their key innovation is a geometric insight that transforms the complex task of calculating this score into a simple computational geometry problem: finding the lower convex hull of a set of 2D points.
This allows the score to be computed in fast time, making the method practical even for problems with thousands of classes.
Experiments on image classification and large language models show that SOCOP significantly increases the rate of singleton predictions (often by over 20%) with only a minimal cost to the average set size, proving its effectiveness and practical value.

**Strengths:**

*   The paper's main strength lies in its novel objective function and the corresponding insight into evaluating conformal prediction. By shifting the focus from minimizing average set size to minimizing the probability of non-singleton sets (`P(size > 1)`), the work introduces a practical and well-motivated criterion for efficiency that directly relates to the goal of automated decision-making. This reframing provides a valuable perspective on what constitutes a "useful" prediction set in many real-world applications.

*   This conceptual contribution is supported by a thorough and well-explained computational algorithm. The paper provides a deep discussion on how to make the proposed nonconformity score practical, culminating in an efficient O(K) method based on a geometric reformulation involving the lower convex hull. This ensures the method is not just a theoretical construct but a scalable tool that can be readily applied.

*   The work can be seen as a natural and timely extension of recent research on improving the efficiency of conformal sets, such as the work on length optimization by [1]. While that paper focused on minimizing the expected set size, this paper explores a related but distinct objective—maximizing singleton frequency. This progression shows a thoughtful engagement with the state-of-the-art and pushes the conversation forward on defining and optimizing different notions of efficiency in conformal prediction.


[Reference]

[1] Kiyani, Shayan, George J. Pappas, and Hamed Hassani. 2024. “Length Optimization in Conformal Prediction.” In Neural Information Processing Systems. Vol. 37. Curran Associates, Inc.

**Weaknesses:**

*   The practical significance of the proposed objective could be further elaborated. While the goal of maximizing singletons is intuitive, the paper could benefit from a more direct discussion on the distinction between the method's singleton predictions and simply using the standard top-1 prediction. A clearer explanation of the trade-offs would help readers understand when the statistical guarantees of a conformal singleton justify the method's complexity, especially since it can still produce non-singleton sets.

*   The method introduces a hyperparameter, $\lambda$, that requires tuning, which can be a practical challenge. The paper could provide a more in-depth discussion on the sensitivity of the results to the choice of $\lambda$. It might also be beneficial to explore the potential impact of tuning bias on the final performance, a topic investigated in recent work such as [1], and consider if more principled selection methods could be developed.

*   The clarity of the writing in the methodology section could be improved. While the experimental results are presented clearly, the section explaining the method itself consists of large blocks of text that can make it difficult to follow the main ideas. The presentation would be enhanced by breaking down the key concepts more, perhaps by using more structured text or adding a high-level diagram to help illustrate the geometric approach.

[Reference]

[2] Zeng, Hao, Kangdao Liu, Bingyi Jing, and Hongxin Wei. 2025. “Parametric Scaling Law of Tuning Bias in Conformal Prediction.” In Forty-Second International Conference on Machine Learning.

**Questions:**

1.  Could you please make the practical difference between a singleton set from your method and a standard top-1 prediction? For instance, in what specific scenarios would a decision-maker prefer a SOCOP singleton over a top-1 prediction, especially considering SOCOP still produces multi-sets in other cases? A more direct comparison of their respective risk profiles would be helpful.

2.  The performance of SOCOP depends on the hyperparameter $\lambda$, which is selected on a tuning set. How sensitive is the method's performance to the choice of $\lambda$ and the size of the tuning set? Could you comment on the potential for tuning bias, as discussed in works like [2], and perhaps provide practical guidance for users on how to select $\lambda$ robustly?

3.  Could you share your thoughts on potential future directions for this work? For example, could this singleton-optimization framework be extended to other, more complex conformal prediction methods beyond the split-conformal setting, such as those that aim for conditional coverage?

4. For theorem part, is there any insight for the P(size <1)'s increasing under proposed method?

---

> ### Author Response · Authors · 2025-11-19
>
> ## Weaknesses
>
> > **Weakness 1**  The practical significance of the proposed objective could be further elaborated. While the goal of maximizing singletons is intuitive, the paper could benefit from a more direct discussion on the distinction between the method's singleton predictions and simply using the standard top-1 prediction. A clearer explanation of the trade-offs would help readers understand when the statistical guarantees of a conformal singleton justify the method's complexity, especially since it can still produce non-singleton sets.
>
> We thank the reviewer for this insightful comment. The key distinction between our SOCOP and the standard top-1 predictor lies in the statistical guarantee. A top-1 prediction has no statistical guarantee. Specifically, if we have a relatively inaccurate classifier, for instance, one that achieves only 70\% accuracy, then the top-one predictions will not achieve a high level of coverage/correctness, such as 95\%. Of course, one could instead output top-$k$ sets until the top-$k$ accuracy reaches 95%, possibly selecting $k$ via conformal prediction.
>
> However, this naive approach outputs a uniformly sized and potentially large prediction set for every individual input. It is more desirable to output small prediction sets for at least some of the inputs. This motivates conformal prediction with more general nonconformity scores, such as $\hat{p}(Y|X)$. However, these classical nonconformity scores target other objectives, such as minimizing expected set size, rather than maximizing singleton predictions.
>
> In contrast, our SOCOP score is optimized to increase the probability of producing singleton predictions while ensuring that on average the true label is contained in the prediction set with probability at least $1-\alpha$. This requires formulating and solving a new optimization problem to design a new non-conformity score and then leveraging an appropriate version of conformal prediction (split conformal prediction in our case) in order to obtain finite sample guarantees.
>
> > **Weakness 2** The method introduces a hyperparameter, $\lambda$, that requires tuning, which can be a practical challenge. The paper could provide a more in-depth discussion on the sensitivity of the results to the choice of $\lambda$. It might also be beneficial to explore the potential impact of tuning bias on the final performance, a topic investigated in recent work such as [1], and consider if more principled selection methods could be developed.
>
> We thank the reviewer for raising these important practical considerations. In fact, **our paper already provides a detailed analysis of $\lambda$'s role and effect**. We have provided a detailed sensitivity analysis of the effect of $\lambda$ in Section 3.1.1 and Figure 2, demonstrating that it acts as a smooth control parameter for the trade-off between average set size and non-singleton probability. The experiments demonstrate that the performance is stable to small changes in $\lambda$, but meanwhile, the model performance can be steered smoothly and reliably with larger changes in $\lambda$ as desired. Practitioners can choose $\lambda$ according to their own preference by drawing the tradeoff plot (the right panel of Figure 2) on their tuning dataset. For full analysis, we provide comprehensive tables in the Appendix showing performance across a variety of $\lambda$ values (Tables 5-9 for ImageNet-Val and Tables 11-15 for ImageNet-V2). We hope that this detailed analysis provides sufficient in-depth discussion to the behavior and role of $\lambda$. If there are any further specific questions you would like to ask, we are happy to investigate.
>
> Second, **our experimental design protocol explicitly avoids the "tuning bias"** the reviewer mentions. We recognize that, in general, data-dependent hyperparameter selection can introduce tuning bias in finite-sample coverage, as discussed by [1]. However, our protocol explicitly avoids this by using a standard three-way data split (tuning, calibration, and evaluation). The hyperparameter $\lambda$ is selected only on the fresh tuning subset. This also enables principled selection via trade-off curve on the tuning set (Figure 2) or automated methods such as the kneedle algorithm like we did. As demonstrated by the broad range of experiments that we present, this approach is practically effective and leads to improved performance compared to a variety of baselines. In future work, however, it might be of interest to study how one could potentially select the hyperparameter $\lambda$ using the calibration dataset and what tuning bias it might introduce, especially for settings where the available data is small so that data reuse can be of value.
>
> We will update our paper to discuss this.
>
> [1] Zeng, Hao, Kangdao Liu, Bingyi Jing, and Hongxin Wei. 2025. “Parametric Scaling Law of Tuning Bias in Conformal Prediction.” In Forty-Second International Conference on Machine Learning.

---

> ### Author Response · Authors · 2025-11-19
>
> > **Weakness 3** The clarity of the writing in the methodology section could be improved. While the experimental results are presented clearly, the section explaining the method itself consists of large blocks of text that can make it difficult to follow the main ideas. The presentation would be enhanced by breaking down the key concepts more, perhaps by using more structured text or adding a high-level diagram to help illustrate the geometric approach.
>
> Thank you for your suggestion to improve the presentation. While some other reviewers already found the presentation of the paper to be adequate, it may certainly be possible to improve it further.
> In the revision, we will use our additional page to expand the key concepts more and use more structured text in the section explaining the method itself.
>
> ## Questions
>
> > **Question 1** Could you please make the practical difference between a singleton set from your method and a standard top-1 prediction? For instance, in what specific scenarios would a decision-maker prefer a SOCOP singleton over a top-1 prediction, especially considering SOCOP still produces multi-sets in other cases? A more direct comparison of their respective risk profiles would be helpful.
>
> Since this question seems to be largely overlapping with  Weakness 1, we would like to ask you to please refer to our response to that question.
>
> To elaborate, a decision maker would prefer SOCOP to top-1 predictions when they desire guaranteed finite sample coverage of the true class label, as it is provided by conformal prediction. For instance, in certain safety-critical settings, such as medical applications, it might be important to be 99\% certain that the true answer is included in the ones that we are considering. In such settings, decision makers should use conformal prediction-type techniques, including ours, to guarantee finite sample validity.
>
> > **Question 2** The performance of SOCOP depends on the hyperparameter $\lambda$, which is selected on a tuning set. How sensitive is the method's performance to the choice of $\lambda$ and the size of the tuning set? Could you comment on the potential for tuning bias, as discussed in works like [1], and perhaps provide practical guidance for users on how to select $\lambda$ robustly?
>
> Since this question seems to be largely overlapping with  Weakness 2, we would like to ask you to please
> refer to our response to that question.
>
> > **Question 3**  Could you share your thoughts on potential future directions for this work? For example, could this singleton-optimization framework be extended to other, more complex conformal prediction methods beyond the split-conformal setting, such as those that aim for conditional coverage?
>
> Thank you for this insightful suggestion. Our proposed SOCOP non-conformity score is designed to be modular. While our paper implements it using standard split conformal prediction, the score itself can be seamlessly integrated into existing frameworks that aim for conditional coverage. For example, our SOCOP score $r(x,y)$ could be directly used as the input non-conformity score $S(x,y)$ in recent work [2]. Specifically, our SOCOP score $r(x,y)$ can be plugged into their augmented quantile regression objective (their Equation (2.4)). The resulting prediction set (defined by their Equation (2.5)) would then aim to satisfy approximate conditional coverage (Theorem 2), while simultaneously inheriting SOCOP's optimization goal of prioritizing singleton sets.
>
> Moreover, another research question could be if one can design a SOCOP-like nonconformity score that is directly aware and targeted for conditional coverage. This might require rethinking the foundations and derivation of SOCOP from the beginning and retracing the steps that we have taken but with a conditional aware optimization objective such as the one from [2]. This direction could potentially be challenging, but it remains interesting for future work.
>
> [2]  Isaac Gibbs, John J Cherian, and Emmanuel J Cand`es. Conformal prediction with conditional guarantees. Journal of the Royal Statistical Society Series B: Statistical Methodology, page qkaf008, 2025.

---

> ### Author Response · Authors · 2025-11-19
>
> > **Question 4**  For theorem part, is there any insight for the $P(size <1)$'s increasing under proposed method?
>
> We thank the reviewer for this insightful question. In the classification task,  $\text{size}<1$ corresponds to empty sets. When $\lambda=0$ (pure singleton limit, see Section 2.2, Corollary 2.5), the resulting split conformal prediction sets are either the top-1 label $\left\\{y_1\right\\}$ if $\hat{p}\left(y_1 \mid x\right) \geqslant c$, for some quantity $c$, or the whole set $\mathcal{Y}$ otherwise. Hence, the procedure will never produce empty sets. However, we agree that when $\lambda>0$, the method can produce empty sets. In our construction, the prediction set at a test point $x$ is $\widehat{C}(x)=\left\\{y: r_\lambda(x, y) \leq \hat{q}\right\\}$, where $\hat{q}$ is the calibrated threshold. Because the scores are non-decreasing, i.e., $r_\lambda(x, y_1)\leq r_\lambda(x, y_2)\leq...\leq r_\lambda(x, y_K)$, $\widehat{C}(x)=\varnothing$ is equivalent to $r_\lambda(x, y_1)>\hat{q}$. One can show that $r_\lambda\left(x, y_1\right)=\frac{\lambda}{\hat{p}\left(y_1 \mid x\right)}$, hence $\widehat{C}(x)=\varnothing$ if and only if $\hat{p}\left(y_1 \mid x\right)<\frac{\lambda}{\hat{q}}$. When the model is very uncertain about an input x, the probability of its top prediction, $\hat{p}(y_1|x)$, can be lower than $\frac{\lambda}{\hat{q}}$, resulting an empty set. In such cases, SOCOP produces an empty set, which can be viewed as the model appropriately abstaining from prediction due to low confidence.
>
> Empirically, the rate of empty sets for SOCOP was negligible ($<0.01\%$) across all experiments, which was comparable to other baseline methods. As this metric did not reveal a meaningful difference, we omitted it from our result tables for conciseness. This confirms that while the theoretical possibility exists, it does not negatively impact the utility of the method in standard classification tasks.

---

> > ### Comment · Reviewer_67Nw · 2025-11-26
> >
> > Thank you for the response. My concerns have been mainly resolved, and I raise the rating. There are some other suggestions:
> >
> > -- The structure of the sections could still be further refined; currently, it is organized into three large sections. Introducing a more fine-grained structure would greatly improve readability and help guide the reader through the key ideas.
> >
> > -- If a revised PDF is uploaded, it would be helpful to highlight newly added or modified content in blue for ease of review.

---

> > > ### Author Response · Authors · 2025-11-26
> > >
> > > Thank you so much for your support of this work! We are grateful for your valuable expertise. We have updated the paper with changes highlighted in blue. Regarding your helpful comment that the structure of the paper could be further refined, we fully agree that a more fine-grained organization can improve readability. We restructured Section 2 by introducing two explicit subsections to guide the reader through the methodology: **Section 2.1 Problem Formulation** and **Section 2.2  Definition of the Nonconformity Score**, followed by the algorithmic details in Section 2.3. If you have other specific suggestions, we are happy to  incorporate them into our paper.
> > >
> > > Sincerely, The authors

---

> > > > ### Comment · Reviewer_67Nw · 2025-11-26
> > > >
> > > > Thank you for your response. There are no further suggestions at this time

---

### Official Review · Reviewer_b3RD · 2025-10-31

**Soundness:** 3
**Presentation:** 3
**Contribution:** 2
**Rating:** 4
**Confidence:** 4

**Summary:**

The paper targets increasing the probability that conformal prediction sets are singletons (sets of size one) in multi-class classification. It proposes a new nonconformity score derived from a Lagrangian relaxation of a joint objective balancing singleton frequency and expected set size, and introduces an O(K) per-instance algorithm via a lower convex hull construction. Empirically, the method increases singleton frequency with small impact on average set size while maintaining marginal coverage.

**Strengths:**

Clarity and organization: The paper is well written and easy to follow. The problem setup, the Lagrangian relaxation, and the geometric algorithm are clearly presented.

Novel scoring and efficient computation: The nonconformity score is new and is explicitly motivated by the singleton objective rather than average size alone. The geometric O(K) algorithm based on the lower convex hull is elegant and practically important.

**Weaknesses:**

1. The statement that “often the most desirable outcome is an unambiguous prediction, a singleton set containing only one label” needs stronger justification. Conformal prediction naturally returns sets of varying sizes; prioritizing singletons is application-dependent rather than universally optimal. Please provide concrete justification and cite papers explicitly values singleton sets.

2. Compared with APS [1], the method provides marginal but not conditional validity when the true $p(y\mid x)$ is given.

3. While the score is motivated by minimizing $P(|C(X)|>1) + \lambda E[|C(X)|]$, similar composite losses can potentially be handled within existing efficiency-oriented frameworks (e.g., [2]). Please clarify why a new score is necessary rather than adapting losses within those frameworks, and whether SOCOP can be integrated into them while preserving nestedness, monotonicity, and computational benefits.

[1] Romano, Yaniv, Matteo Sesia, and Emmanuel Candès. Classification with valid and adaptive coverage. NeurIPS 33 (2020): 3581–3591.

[2] Liang, Ruiting, Wanrong Zhu, and Rina Foygel Barber. Conformal prediction after efficiency-oriented model selection. arXiv:2408.07066 (2024).

**Questions:**

The definition of a singleton set should be more precise. The definition used in the formulas is $|C(X)| \le 1$, which differs from $|C(X)| = 1$. Empty prediction sets can occur in conformal prediction. For example, if $\hat p$ can almost perfectly predict the true class, then approximately an $\alpha$ fraction of least ambiguous sets will be empty. If we use $|C(X)| = 1$ as the definition of a singleton set and change the first term of the loss in line 135 to $I(|C(X)| \ne 1)$, this may harm the monotonicity of the score function.

---

> ### Author Response · Authors · 2025-11-19
>
> ## Weaknesses
>
> > **Weakness 1.** The statement that “often the most desirable outcome is an unambiguous prediction, a singleton set containing only one label” needs stronger justification. Conformal prediction naturally returns sets of varying sizes; prioritizing singletons is application-dependent rather than universally optimal. Please provide concrete justification and cite papers explicitly values singleton sets.
>
> Thank you for this question. We agree with the reviewer that prioritizing singleton sets is application-dependent. As we have already described in the paper l045, this is "often" the case, and we have not claimed that it is universally true. Our motivation comes from practical scenarios where an unambiguous prediction (a singleton) is operationally more valuable than an ambiguous one. A concrete example is in medical diagnostics, such as the classification of tissue samples (which we evaluate in our TissueMNIST experiment, Sec. 3.2). As we briefly state in the introduction (Page 1, Paragraph 3), our motivation stems from downstream workflows where a set of size two or more may require additional human intervention and thus brings an outsized cost.
>
> This motivation aligns with recent studies that explicitly value singletons in downstream decision processes. In particular, [1] note that a key drawback of set-valued predictions is that “decision making in the real world often requires us to act on a single option”, which conformal methods do not always guarantee. Their human user study further showed that human accuracy was highest when the conformal prediction set was a singleton, and it dropped as the set size grew. These findings support our design goal: while marginal coverage ensures reliability, optimizing for singleton frequency enhances practical usability in domains where a single confident prediction is operationally preferred.
>
> In addition, as we discuss in Section 2.2 (``Extension to $P(\text{size}>k_0)$"), our framework is not fixed to $k_0=1$. It naturally extends to control the probability of sets larger than any threshold $k_0$ (e.g., $k_0=2, 3, \text{or } 10$). This flexibility allows practitioners to choose the optimization criterion based on their specific needs.
>
> [1] Jesse C Cresswell, Yi Sui, Bhargava Kumar, and No¨el Vouitsis. Conformal prediction sets improve human
> decision making. arXiv preprint arXiv:2401.13744, 2024.

---

> ### Author Response · Authors · 2025-11-19
>
> > **Weakness 2.** Compared with APS [2], the method provides marginal but not conditional validity when the true $p(y|x)$ is given.
>
> We thank the reviewer for highlighting this important point. We agree with the reviewer that when the true $p(y|x)$ is given, the simplest way to achieve conditional validity is APS [2], which includes classes from highest to lowest probability
> until their sum just exceeds the threshold. However, in real machine learning tasks, we do not have access to such an oracle. APS [2] also uses split conformal prediction to achieve marginal coverage.
> To be clear, in practice, their method does not achieve conditional coverage.
> Therefore, we do not necessarily believe that achieving conditional coverage, when the true probabilities are given, is the most important goal, because, in reality, we always have to face scenarios where the conditional probabilities are not given. Thus, it is more meaningful to study criteria for this setting.
>
> There are some other frameworks that aim to provide conditional guarantees, even when the true probabilities are not known, such as [3].
> To approximately achieve conditional coverage, one can also incorporate our non-conformity score into these existing frameworks. Specifically, our SOCOP score $r(x,y)$ can be plugged into their augmented quantile regression objective (their Equation (2.4)). The resulting prediction set (defined by their Equation (2.5)) would then aim to satisfy approximate conditional coverage (Theorem 2), while simultaneously inheriting SOCOP's optimization goal of prioritizing singleton sets.
>
> In addition, thanks to Reviewer 5adu's suggestion, we also add new evaluations on size-stratified coverage violation (SSCV), which is introduced in RAPS [4] as a measure of adaptiveness and conditional coverage violation. Following [4], we use the same set-size strata and tune the hyperparameter $\lambda$ to minimize SSCV on the tuning set. The results of Resnet152-v2 and ViT-h-14 on ImageNet-Val are:
> | Method         | Coverage     | Avg Size       | P(Size>1)    | SSCV         |
> | -------------- | ------------ | -------------- | ------------ | ------------ |
> | Plug-in        | 0.969 ±0.003 | 47.362 ±7.138  | 0.469 ±0.025 | 0.046 ±0.001 |
> | RAPS           | 0.950 ±0.002 | 8.568 ±1.580   | 0.448 ±0.012 | 0.031 ±0.011 |
> | Pure Singleton | 0.950 ±0.002 | 250.539 ±4.554 | 0.250 ±0.005 | 0.050 ±0.000 |
> | LAS            | 0.950 ±0.002 | 2.279 ±0.046   | 0.467 ±0.007 | 0.197 ±0.026 |
> | SOCOP (ours)   | 0.950 ±0.002 | 3.372 ±0.198   | 0.304 ±0.008 | 0.039 ±0.009 |
>
> | Method         | Coverage     | Avg Size       | P(Size>1)    | SSCV         |
> | -------------- | ------------ | -------------- | ------------ | ------------ |
> | Plug-in        | 0.976 ±0.001 | 8.529 ±0.805   | 0.356 ±0.008 | 0.048 ±0.002 |
> | RAPS           | 0.950 ±0.002 | 7.652 ±2.259   | 0.319 ±0.007 | 0.047 ±0.003 |
> | Pure Singleton | 0.950 ±0.003 | 136.219 ±4.980 | 0.135 ±0.005 | 0.050 ±0.000 |
> | LAS            | 0.950 ±0.002 | 1.291 ±0.011   | 0.224 ±0.006 | 0.126 ±0.119 |
> | SOCOP (ours)   | 0.950 ±0.002 | 1.519 ±0.068   | 0.155 ±0.006 | 0.041 ±0.016 |
>
> Results are averaged over 100 random data splits. Please note that for the SSCV evaluation, we followed the protocol in [4] by selecting $\lambda$ from a grid of small values (specifically, smaller than the candidates used to optimize average size) to minimize SSCV on the tuning set. It is worth noting that while [4] reports these optimized SSCV figures, they do not report the corresponding average set sizes, which are typically much larger under this specific optimization setting (increases from $\approx3.2$ to $\approx8.6$ for Resnet152-v2 and from $\approx1.4$ to $\approx7.7$ for ViT-h-14).
>
> In contrast, **our SOCOP method achieves a comparable SSCV while maintaining a reasonably small average size and a significantly lower non-singleton probability**, demonstrating that SOCOP can achieve adaptivity without sacrificing efficiency.
>
> [2] Romano, Yaniv, Matteo Sesia, and Emmanuel Candès. Classification with valid and adaptive coverage. NeurIPS 33 (2020): 3581–3591.
>
> [3] Isaac Gibbs, John J Cherian, and Emmanuel J Cand`es. Conformal prediction with conditional guarantees. Journal of the Royal Statistical Society Series B: Statistical Methodology, page qkaf008, 2025.
>
> [4] Anastasios Nikolas Angelopoulos, Stephen Bates, Michael Jordan, and Jitendra Malik. Uncertainty sets for image classifiers using conformal prediction. In International Conference on Learning Representations, 2021. URL https://openreview.net/forum?id=eNdiU_DbM9.

---

> ### Author Response · Authors · 2025-11-19
>
> > **Weakness 3.**  While the score is motivated by minimizing $P(|C(X)|>1) + \lambda E(|C(X)|)$, similar composite losses can potentially be handled within existing efficiency-oriented frameworks (e.g., [5]) . Please clarify why a new score is necessary rather than adapting losses within those frameworks, and whether SOCOP can be integrated into them while preserving nestedness, monotonicity, and computational benefits.
>
> We thank the reviewer for this insightful question. The efficiency-oriented frameworks in [5, 6] has a core difference from ours. As the reviewer suggested, if we use our composite loss $P(|C(X)|>1) + \lambda E(|C(X)|)$ in the efficiency-oriented framework, it actually uses this loss as a metric to select a score from pre-specified candidates $\Lambda = \\{S^1, S^2, \dots\\}$. This is fundamentally different from directly optimizing a loss to derive a score (our work). **Thus, we do not think that the two approaches aim to solve the same problem, making them not directly comparable in our setting.** For example, take $\Lambda=\\{S^{LAS}, S^{APS}, S^{RAPS}\\}$, (here we assume we have enough data to split into tuning/calibration/test sets to avoid selection bias, like we did in our experiments).  The best score on the tuning set will determine the performance on the test set, and will be exactly that of the corresponding baseline/score reported in our tables.
>
> In other words, the loss is only a selection metric among several candidate scores, and does not fundamentally improve them.
> In contrast, in our case, the loss determines the construction of the score, which can potentially lead to an improved performance compared to a fixed set of baseline scores. In addition, integrating our loss into [5] would require first tuning $\lambda$ beofre proceeding to select the score. This would introduce the additional difficulty of selecting the hyperparameter $\lambda$, and their framework would have to be potentially extended to allow the selection of not just non-conformity scores but also loss functions. (At the outset, at least, their loss function is fixed and pre-specified.)
>
> For the second question, in principle, our SOCOP can be integrated into those frameworks while preserving nestedness, monotonicity, and computational benefits, as these properties are only related to the score computation. However, our score also depends on $\lambda$. One possible way to deal with this challenge is to fix the same $\lambda$ for both the score and the loss, i.e., $\Lambda=\\{S^{SOCOP}_\lambda, \text{other scores}\\}$ with the selection loss $\mathcal{L}=P(|C(X)|>1) + \lambda E(|C(X)|)$, but this again goes back to the above hyperparameter selection problem. In contrast, our work provides a flexible way to choose the $\lambda$ based on the tradeoff (Figure 2, Sec. 3.1.1) between size and multiple labels rate. Practitioners can choose the $\lambda$ based on their own preference.
>
> We will add this discussion to our paper.
>
> > **Question 1.** The definition of a singleton set should be more precise. The definition used in the formulas is $|C(X)|\leq 1$
> , which differs from $|C(X)|= 1$. Empty prediction sets can occur in conformal prediction. For example, if $\hat{p}$ can almost perfectly predict the true class, then approximately an $\alpha$ fraction of least ambiguous sets will be empty. If we use $|C(X)|= 1$ as the definition of a singleton set and change the first term of the loss in line 135 to $I(|C(X)|\neq 1)$, this may harm the monotonicity of the score function.
>
> We thank the reviewer for this key observation. We agree that a singleton set should precisely refer to $|C(X)|=1$.
> Our use of the "singleton objective" was imprecise. This objective is formally known as the M-criterion (for multiple labels) from [7]. Since we found the term M-criterion a bit too terse and non-descriptive, we decided to choose the more descriptive term "singleton objective"; However, we agree that its meaning needs to be more precise. We will add a clarification about the precise meaning of singletons, but if the reviewer agrees with us, we would like to keep the current terminology. Please let us know, and we are happy to discuss further.
>
> Across all datasets used in our paper, we observe that empty prediction sets occur in fewer than 0.01\% of test instances for all methods, except for RAPS (roughly 0.1\%). Hence, we do not report this metric in the text as its effect is completely negligible. However, in the revision, we will add a footnote about this issue.
>
> [5] Ruiting Liang, Wanrong Zhu, and Rina Foygel Barber. Conformal prediction after efficiency-oriented
> model selection. arXiv preprint arXiv:2408.07066, 2024.
>
> [6] Yachong Yang and Arun Kumar Kuchibhotla. Selection and aggregation of conformal prediction sets. Journal of the American Statistical Association, 120(549):435–447, 2025.
>
> [7] Vladimir Vovk, Alex Gammerman, and Glenn Shafer. Algorithmic learning in a random world. Springer
> Science & Business Media, 2005.

---

> ### Comment · Reviewer_b3RD · 2025-11-24
>
> I sincerely thank authors' reply. I have the following additional questions.
>
> Q1. Can you also calculate the SSCV for APS? I just realized from your reply to another reviewer that plug-in is not APS.
>
> Q2. Since we want $|C(X)|=1$, will you change $|C(X)|>1$ to $|C(X)|\ne 1$ in the loss function?
>
> Q3. Let me explain how to embed your objective to existing efficiency oriented methods. Your proposed method is very different, but I think it is worth to compare.  A key observation is that if $p(y_2|x)>p(y_1|x)$, then $y_1\in C(x)$ implies $y_2\in C(x)$. I think this is true for most conformal prediction method for classification, including the score function in equation (3) of the present manuscript. If we consider equation (2), we only need to consider $K$ sets for optimization, rather than $2^K$, because we only need to consider top-1 $y$, top-2 $y \dots $ in $p(y|x)$. Given this observation, we can modify the loss in line 136 to a score function.
>
> The loss function contains 3 terms. They corresponds to the following 3 score functions.
>
> (i) First define $rank(x,y) = $ the rank of $p(y|x)$ in $p(y'|x)$ for $y'\in[K]$ ordering from large to small. $S^{SINGLETON}(x,y) = 0$ if $rank(x,y)=1$ and $=1$ if $rank(x,y)>1$. $S^{SINGLETON}(x,y)$ corresponds to the term $I(C(x)>1)$.
>
> (ii) |C(x)| exactly corresponds to $rank(x,y)$. Let's define this as $S^{RANK}(x,y) = rank(x,y)$.
>
> (iii) $-\sum_{y\in C(x)} p(y|x)$ corresponds to $-\sum_{y': p(y'|x)\ge p(y|x)} p(y'|x)$. This is very similar as $S^{APS}$. Let's call this $S^{APS}$.
>
> We define a aggregated score function $S^{AGG}_{\lambda, \eta}(x,y) = S^{SINGLETON}(x,y) + \lambda S^{RANK}(x,y) + \eta S^{APS}(x,y)$. Then we search $(\lambda, \eta)\in \mathbb R^2$ to optimize certain objective in the calibration dataset, such as average prediction set size or proportion of prediction set with more than 1 element. Therefore, unlike [6], the method in [5] or [8] is not limited to the best score out of three, but all possible linear combination of the scores. We only need to change the objective $|C(x)|$ to $|C(x)| + I(|C(x)|>1)$.
>
> In the definition in equation (2), the smallest $S_{\eta, p(\cdot|x), \lambda}$ containing $y$ is $\\{y': S^{AGG}\_{\lambda, \eta}(x,y')\le S_{\lambda, \eta}(x,y)\\}$.
>
> As I mentioned in my review, your proposed score is different from the existing ones like $S^{AGG}_{\lambda, \eta}(x,y)$. You don't use $\ell\_{p(\cdot | x),\lambda}$ directly. You use the Lagrange multiplier of $S^{APS}$ to define the score using equation (3).
>
> Anther framework you should compare with is [9], as mentioned by another reviewer. It is about an efficient oriented approach, but to encourage singleton prediction set, you just need to change the objective.
>
> [8] Luo, Rui, and Zhixin Zhou. "Conformity score averaging for classification." Forty-second International Conference on Machine Learning.
>
> [9] Kiyani, Shayan, George J. Pappas, and Hamed Hassani. 2024. “Length Optimization in Conformal Prediction.” In Neural Information Processing Systems. Vol. 37. Curran Associates, Inc.

---

> > ### Author Response · Authors · 2025-11-25
> >
> > We thank the reviewer for the insightful follow-up questions. We address them in turn below.
> >
> > 1.
> > Yes, the results for APS are:
> >
> > Resnet152-v2:
> > | Method | Coverage        | Avg Size         | P(Size>1)       | SSCV            |
> > |--------|-----------------|------------------|-----------------|-----------------|
> > | APS    | 0.950 ± 0.002   | 46.497 ± 2.110   | 0.423 ± 0.008   | 0.046 ± 0.002   |
> >
> > Vit-h-14:
> > | Method | Coverage        | Avg Size          | P(Size>1)       | SSCV           |
> > |--------|-----------------|-------------------|-----------------|----------------|
> > | APS    | 0.950 ± 0.002   | 10.000 ± 0.786    | 0.318 ± 0.007   | 0.048 ± 0.001  |
> >
> > **RAPS outperforms APS in both Average size and SSCV**, which is consistent with the findings in RAPS paper. As we replied to Reviewer XEKF, we did not include APS as a separate baseline because we consider RAPS as a stronger version of APS.
> >
> >
> > 2.
> > To change $|C(X)|>1$ to $|C(X)|\neq 1$, we would need to go from the very beginning,  rephrasing and replacing  all mathematical results and experiments in our paper completely.
> > As we mentioned in our first reply, we used $|C(X)|>1$ as it originates from the M-criterion in [7]. Since we found the term M-criterion a bit too terse and non-descriptive, we decided to choose the more descriptive term "singleton objective”, though it's not completely precise. We have added a footnote at line 051 to clarify this in the revised version.
> >
> > In addition, thanks to Reviewer 67Nw's question, we also analyzed the probability of empty sets from a theoretical perspective. When $\lambda=0$ (pure singleton limit, see Section 2.2, Corollary 2.5), the resulting split conformal prediction sets are either the top-1 label $\left\\{y_1\right\\}$ if $\hat{p}\left(y_1 \mid x_{n+1}\right) \geqslant c$, for some quantity $c$, or the whole set $\mathcal{Y}$ otherwise. Hence, the procedure will never produce empty sets. However, we agree that when $\lambda>0$, the method can produce empty sets. In our construction, the prediction set at a test point $x$ is $\widehat{C}(x)=\left\\{y: r_\lambda(x, y) \leq \hat{q}\right\\}$, where $\hat{q}$ is the calibrated threshold. Because the scores are non-decreasing, i.e., $r_\lambda(x, y_1)\leq r_\lambda(x, y_2)\leq...\leq r_\lambda(x, y_K)$, $\widehat{C}(x)=\varnothing$ is equivalent to $r_\lambda(x, y_1)>\hat{q}$. One can show that $r_\lambda\left(x, y_1\right)=\frac{\lambda}{\hat{p}\left(y_1 \mid x\right)}$, hence $\widehat{C}(x)=\varnothing$ if and only if $\hat{p}\left(y_1 \mid x\right)<\frac{\lambda}{\hat{q}}$. A typical situation in which this happens is when the model is very uncertain about an input x, the probability of its top prediction, $\hat{p}(y_1|x)$, can be lower than $\frac{\lambda}{\hat{q}}$, resulting in an empty set. In such cases, SOCOP produces an empty set, which can be viewed as the model appropriately abstaining from prediction due to low confidence.
> >
> > Most importantly, **empirically, the rate of empty sets for SOCOP was negligible** (<0.01%) across all experiments, which was comparable to other baseline methods. Thus, in our setting, using $I(|C(X)|>1)$ rather than $I(|C(X)| \neq 1)$ does not affect the conclusions or the practical utility of the method.
> >  Therefore, we do not see a utilitarian justification for going through the effort of changing everything (which may take several weeks of work) because the outcomes do not change in any significant way in the experiments reported in our paper. In short, we do not believe it is reasonable to expect such extensive modifications and a complete rerun of all experiments---especially in a short rebuttal period---given that they do not lead to any valuable outcome for the community.
> >
> >  We hope that our arguments sound reasonable to the reviewer. However, if there is absolutely no way that the reviewer agrees, then we would rather change the name of our method to "Multiplicity Optimized Conformal Prediction" (MOCOP) to ensure consistency of the terminology and objective.

---

> > > ### Author Response · Authors · 2025-11-25
> > >
> > > 3.
> > > **We implemented and tested the aggregated score $S_\{\lambda,\eta\}^\{\mathrm{AGG}\}$
> > > proposed by the reviewer.** This score requires hyperparameter tuning of $(\lambda,\eta)$, which introduces the difficulty of choosing loss functions to trade off average size and singleton probability (an issue we discussed in our first response). To fully evaluate the performance of the score the reviewer suggested, we consider three losses: $\mathbb{E}[|C(x)|+I(|C(x)|>1]$, $\mathbb{E}[0.1*|C(x)|+I(|C(x)|>1]$ and $\mathbb{E}[I(|C(x)|>1]$, and for each loss, we tune $(\lambda,\eta)$ to minimize the loss in the tuning set.
> > > As the losses are discontinuous and don't have clear structures (such as monotonicity), we didn't find easy ways to optimize for $(\lambda, \eta)$, therefore, we consider grid search over $(\lambda, \eta)$.
> > >  The grid we use is $\{0.002, 0.01, 0.05, 0.25, 1.0\}$ for $\lambda$ and $\{0.001, 0.01, 0.1, 1.0, 10.0\}$ for $\eta$. Since grid search over $(\lambda,\eta)$ is computationally expensive and quite slow, we only average the results over 20 random splits. Here are the results:
> > >
> > > $\mathbb{E}[|C(x)|+I(|C(x)|>1]$:
> > > | Method   | Coverage        | Avg Size          | P(Size>1)       | SSCV            |
> > > |----------|-----------------|-------------------|-----------------|-----------------|
> > > | AGG      | 0.951 ± 0.003   | 4.051 ± 0.152     | 1.000 ± 0.000   | 0.440 ± 0.164   |
> > >
> > > $\mathbb{E}[0.1*|C(x)|+I(|C(x)|>1]$:
> > > | Method   | Coverage        | Avg Size          | P(Size>1)       | SSCV            |
> > > |----------|-----------------|-------------------|-----------------|-----------------|
> > > | AGG      | 0.951 ± 0.003   | 4.130 ± 0.246     | 1.000 ± 0.000   | 0.476 ± 0.200   |
> > >
> > > $\mathbb{E}[I(|C(x)|>1]$:
> > > | Method   | Coverage        | Avg Size           | P(Size>1)       | SSCV            |
> > > |----------|-----------------|--------------------|-----------------|-----------------|
> > > | AGG      | 0.950 ± 0.002   | 14.573 ± 3.207     | 0.961 ± 0.003   | 0.647 ± 0.019   |
> > >
> > > In all three tuning objectives,  **the aggregated score (AGG) either produces prediction sets that are almost never singletons $(P($ size $>1) \approx 1)$ or leads to very large average set sizes, and in all cases yields substantially larger SSCV than SOCOP.** Based on these experiments, we conclude that, in our setting, the proposed aggregated score performs significantly worse than SOCOP in terms of the trade-off between singleton rate, average size, and adaptiveness.
> > >
> > >
> > >  Finally, we thank the reviewer for pointing us to [9]. We already compared against the CPL framework empirically in Appendix D.3, where we treat CPL as a strong length-optimization baseline that is complementary to our singleton-oriented objective. Conceptually, we agree that “encouraging singleton prediction sets” is a natural extension of their framework, and that aligning the CPL objective with ours is an interesting research direction.
> > >
> > > However, technically this is not a matter of simply swapping the loss. The infinite-sample theory of [9] relies crucially on the linearity of the length functional $\mathbb{E}[\text{len}(C(X))]$ in the set indicator $C(x,y)$, which allows them to recast the problem as an (infinite-dimensional) linear program and prove strong duality and conditional guarantees. Replacing length by our “singleton objective” term $I(|C(X)|>1)$ (or $I(|C(X)|>1)+\lambda |C(X)|$) produces a non-convex, discontinuous objective even after the standard $[0,1]$ relaxation of $C$, so the LP structure and strong duality arguments in [9] no longer apply. Extending CPL to handle such non-convex objectives would require reworking their duality and conditional-validity analysis from the ground up, which we view as a substantial and interesting follow-up project rather than a quick modification within the scope of this paper.

---

> ### Comment · Reviewer_b3RD · 2025-11-26
>
> Thanks for your reply and additional experiments. I decide to raise the score to 6.

---

> ### Author Response · Authors · 2025-11-26
>
> Thank you so much for your support of this work! We are grateful for your valuable expertise.
>
> Sincerely,  The authors

---

### Official Review · Reviewer_XEKF · 2025-10-31

**Soundness:** 3
**Presentation:** 3
**Contribution:** 3
**Rating:** 6
**Confidence:** 4

**Summary:**

This paper proposes a new conformal score function designed to produce prediction sets that minimize a weighted sum of the probability of singleton sets and the expected set size, subject to lower bound on marginal coverage. This generalizes the LAC score function proposed in Sadinle et al., 2019, which is a special case of this paper’s score function with the weight on expected set size set to infinity. Computing this score function is nontrivial, and the paper derives a way to compute it efficiently in O(K) time, where K is the number of classes. The method is validated on image and Q&A datasets and we see that the proposed method does a good job balancing average size and P(size > 1).

**Strengths:**

* The paper is very well written. Some parts I found particularly useful: the discussion of connections to Sadinle et al., 2019 and the comparison in Figs 2-3; the motivation for taking a linear combination of the singleton and size objectives starting on line 311
* This method can be easily extended to P(size > $k_0$) for any integer $k_0 \geq 1$. This seems quite practically useful.
* The experiments are extensive and shed insight into the operating characteristics of the proposed method.

**Weaknesses:**

There are some places the writing could be improved. See Questions

**Questions:**

* Can you provide more intuition in the paragraph “Computing the nonconformity score”? My attempt is “The conformal score is the smallest “price” $\eta$ in terms of the set size metric (singleton + size objective) per unit of “coverage” such that if we “buy” all classes with price less than $\eta$, we will have purchased the true class. I'm not sure if this is precisely correct, but something like this would be useful.
* Why do you use Plug-In sets (which do not have a marginal coverage guarantee) instead of APS (which is essentially the same procedure but with the threshold tuned to achieve a marginal coverage guarantee)?
* “Our method SOCOP outperforms Plug-In and RAPS in both Average Size and P(size > 1). We find this result remarkable […]” — I don’t think the second sentence is warranted. Plug-In/APS and RAPS target X-conditional coverage. How does SOCOP do on this metric?
* On line 382, you say that ImageNet-V2 is more challenging test dataset. Can you briefly describe why in the text?
* A class-conditional extension of this method that is mentioned in the Discussion section seems quite important to me, as there are many classification tasks where class-conditional coverage is important. However, I don’t immediately see a way that the current methodology could be extended. Do you have thoughts on how?


Small corrections/suggestions:
* Line 162: “result;” -> “result,”
* Line 200: “that, for k >= 3, that” -> “that, for k >= 3,”
* Line 231: “the the jumps”
* Line 267: “is set” -> “is the set”
* Line 337: “Results are reported” -> “Results for this method are reported” (initially I thought this sentence was referring to results for all methods, not just CPL)
* Line 365: “Imagenet” -> “ImageNet”
* Line 399: Fig 2 caption “for of”

---

> ### Author Response · Authors · 2025-11-19
>
> ## Questions
>
> > **Question 1.** Can you provide more intuition in the paragraph “Computing the nonconformity score”? My attempt is “The conformal score is the smallest “price” $\eta$ in terms of the set size metric (singleton + size objective) per unit of “coverage” such that if we “buy” all classes with price less than $\eta$, we will have purchased the true class. I'm not sure if this is precisely correct, but something like this would be useful.
>
> We thank the reviewer for this insightful question and intuition. Indeed, the proposed intuition captures the essence of our construction. Our nonconformity score $r(x,y)$ is derived from the Lagrangian objective $\ell_{\gamma,\lambda}(S;\eta) = \mathbb{I}(|S|>1) + \lambda|S|-\eta   \sum_{y \in S} \gamma_y$. To clarify this intuition, define $\operatorname{Cost}(S) = \mathbb{I}(|S|>1) + \lambda|S|$ as the cost of penalizing non-singleton sets, and $\operatorname{Benefit}(S) = \sum_{y \in S} \gamma_y$ as the benefit of covered probability mass. The multiplier $\eta\geq0$ can indeed be viewed as the price per unit of benefit (coverage).
>
> For each fixed $\eta$, the optimal set $S_{\eta, \gamma} \in \arg\min_{S \subseteq \mathcal{Y}} \ell_{\gamma,\lambda}(S;\eta)$ minimizes the net cost given this "price" $\eta$. As $\eta$ increases, the optimization is incentivized to "purchase" more coverage, leading to nested sets $S_{\eta_1, \gamma} \subseteq S_{\eta_2, \gamma}$ for $\eta_1<\eta_2$ (Lemma B.1). Our score, $r(x,y) := \inf\\{\eta \ge 0 : y \in S_{\eta,\gamma}\\}$, thus quantifies the entry price at which the true label $y$ is first included in the optimal set.
>
> To summarize, your interpretation is correct: the score $r(x, y_{true})$ is the minimum "price" $\eta$ required to "purchase" the true class. We will add this clarifying intuition to our revised version.
>
> > **Question 2.** Why do you use Plug-In sets (which do not have a marginal coverage guarantee) instead of APS (which is essentially the same procedure but with the threshold tuned to achieve a marginal coverage guarantee)?
>
> Thank you for this point. The term Plug-In sets in our paper follows the definition of the naive prediction sets from [1], which includes classes from highest to lowest probability until their sum just exceeds the threshold $1-\alpha$. This method serves as a non-conformal baseline.
>
> You are correct that APS is the same procedure but with conformal calibration. We did not include APS as a separate baseline because RAPS is a stronger version of APS for set-size efficiency. Moreover, APS is a special case of RAPS with $\lambda=0$, and in our experiments with ImageNet, we observed that $\lambda=0$ was never selected, confirming that RAPS outperforms APS. So we omitted APS, but we are happy to include it as a separate baseline if you prefer. (Though we would appreciate a justification of why to do this.)
>
> [1] Anastasios Nikolas Angelopoulos, Stephen Bates, Michael Jordan, and Jitendra Malik. Uncertainty sets for image classifiers using conformal prediction. In International Conference on Learning Representations, 2021. URL https://openreview.net/forum?id=eNdiU_DbM9.

---

> ### Author Response · Authors · 2025-11-19
>
> > **Question 3.** “Our method SOCOP outperforms Plug-In and RAPS in both Average Size and $P(size > 1)$. We find this result remarkable […]” — I don’t think the second sentence is warranted. Plug-In/APS and RAPS target $X$-conditional coverage. How does SOCOP do on this metric?
>
> We thank the reviewer for this insightful comment. First of all, we want to clarify that both the APS/RAPS paper and our paper focus on the standard split-conformal framework that targets marginal coverage. APS/RAPS aims to improve set size efficiency, while our SOCOP targets the combination of set size efficiency and non-singletion rate. Our sentence "We find this result remarkable" refers to marginal validity. We agree that APS is derived from a conditional perspective, but, ultimately, it does not have a finite-sample conditional validity, as it is used with a learned score. Thus, **we believe that marginal validity is a reasonable metric to evaluate and compare methods with**. However, we agree that our claim is perhaps a bit exaggerated, and we would be happy to tone it down in the revision.
>
> To evaluate the performance of conditional coverage and adaptiveness,  APS [2] introduces Worst-slice coverage (WSC), and RAPS [1] introduces size-stratified coverage violation (SSCV). Since the WSC metric requires searching for the ``worst” data slice $S_{v,a,b} = \\{x : a \le v^T X \le b\\}$, which is computationally intractable for very high-dimensional feature spaces such as those for ImageNet in our paper, we choose to evaluate the SSCV metric. Following [1], we use the same set-size strata and tune the hyperparameter $\lambda$ to minimize SSCV on tuning set. The results of Resnet152-v2 and ViT-h-14 on ImageNet-Val are:
> | Method         | Coverage     | Avg Size       | P(Size>1)    | SSCV         |
> | -------------- | ------------ | -------------- | ------------ | ------------ |
> | Plug-in        | 0.969 ±0.003 | 47.362 ±7.138  | 0.469 ±0.025 | 0.046 ±0.001 |
> | RAPS           | 0.950 ±0.002 | 8.568 ±1.580   | 0.448 ±0.012 | 0.031 ±0.011 |
> | Pure Singleton | 0.950 ±0.002 | 250.539 ±4.554 | 0.250 ±0.005 | 0.050 ±0.000 |
> | LAS            | 0.950 ±0.002 | 2.279 ±0.046   | 0.467 ±0.007 | 0.197 ±0.026 |
> | SOCOP (ours)   | 0.950 ±0.002 | 3.372 ±0.198   | 0.304 ±0.008 | 0.039 ±0.009 |
>
> | Method         | Coverage     | Avg Size       | P(Size>1)    | SSCV         |
> | -------------- | ------------ | -------------- | ------------ | ------------ |
> | Plug-in        | 0.976 ±0.001 | 8.529 ±0.805   | 0.356 ±0.008 | 0.048 ±0.002 |
> | RAPS           | 0.950 ±0.002 | 7.652 ±2.259   | 0.319 ±0.007 | 0.047 ±0.003 |
> | Pure Singleton | 0.950 ±0.003 | 136.219 ±4.980 | 0.135 ±0.005 | 0.050 ±0.000 |
> | LAS            | 0.950 ±0.002 | 1.291 ±0.011   | 0.224 ±0.006 | 0.126 ±0.119 |
> | SOCOP (ours)   | 0.950 ±0.002 | 1.519 ±0.068   | 0.155 ±0.006 | 0.041 ±0.016 |
>
> Results are averaged over 100 random data splits. Please note that for the SSCV evaluation, we followed the protocol in [1] by selecting $\lambda$ from a grid of small values (specifically, smaller than the candidates used to optimize average size) to minimize SSCV on the tuning set. It is worth noting that while [1] reports these optimized SSCV figures, they do not report the corresponding average set sizes, which are typically much larger under this specific optimization setting (increases from $\approx3.2$ to $\approx8.6$ for Resnet152-v2 and from $\approx1.4$ to $\approx7.7$ for ViT-h-14).
>
> In contrast, **our SOCOP method achieves a comparable SSCV while maintaining a reasonably small average size and a significantly lower non-singleton probability**, demonstrating that SOCOP can achieve adaptivity without sacrificing efficiency.
>
> Finally, both our score and APS/RAPS score are modular, which can be seamlessly plugged into the existing framework that aims to approximate conditional coverage, like [3].
>
> [2] Yaniv Romano, Matteo Sesia, and Emmanuel Candes. Classification with valid and adaptive coverage. Advances in neural information processing systems, 33:3581–3591, 2020.
>
> [3] Isaac Gibbs, John J Cherian, and Emmanuel J Cand`es. Conformal prediction with conditional guarantees. Journal of the Royal Statistical Society Series B: Statistical Methodology, page qkaf008, 2025.

---

> ### Author Response · Authors · 2025-11-19
>
> > **Question 4.** On line 382, you say that ImageNet-V2 is more challenging test dataset. Can you briefly describe why in the text?
>
> We thank the reviewer for the suggestion. ImageNet-V2 [4] was constructed by re-collecting images through a new sampling pipeline, which results in a natural distribution shift relative to the original training and validation sets. [4] found that a broad range of models has a significant drop in top-1 accuracy on this new dataset. This is consistent with what we observed in our experiments: all five models we evaluated show a clear drop in top-1 accuracy. We will add clarification in our revised version.
>
>
> > **Question 5.** A class-conditional extension of this method that is mentioned in the Discussion section seems quite important to me, as there are many classification tasks where class-conditional coverage is important. However, I don’t immediately see a way that the current methodology could be extended. Do you have thoughts on how?
>
> We thank the reviewer for raising this important point. Indeed, **our SOCOP nonconformity score is modular and can be directly incorporated into the  label-conditional or Mondrian conformal prediction framework** [5]. Specifically, one can perform conformal calibration separately within each class, obtaining class-specific quantiles $\hat{q}\_{1-\alpha, y}$ and forming prediction sets $\hat{C}(x)=\\{y : r(x, y) \le \hat{q}\_{1-\alpha, y}\\}$, thereby guaranteeing class-conditional coverage.
>
> When per-class calibration data are limited,
> the clustered conformal prediction proposed by [6] can be applied. This  first clusters classes based on the similarity of their score distributions and then performs calibration at the cluster level.
>
> Therefore, inserting our SOCOP nonconformity score into Mondrian conformal prediction or clustered conformal prediction can result in prediction sets that satisfy label/cluster-conditional coverage, while simultaneously inheriting SOCOP’s optimization goal of prioritizing singleton sets.
>
> Deriving a version of SOCOP that is label-conditional native is also an interesting future research.
>
> [4] Benjamin Recht, Rebecca Roelofs, Ludwig Schmidt, and Vaishaal Shankar. Do imagenet classifiers
> generalize to imagenet? In International conference on machine learning, pages 5389–5400. PMLR, 2019.
>
> [5] Vladimir Vovk, Alex Gammerman, and Glenn Shafer. Algorithmic learning in a random world. Springer Science & Business Media, 2005.
>
> [6] Tiffany Ding, Anastasios Angelopoulos, Stephen Bates, Michael Jordan, and Ryan J Tibshirani. Class-
> conditional conformal prediction with many classes. Advances in neural information processing systems,
> 36:64555–64576, 2023.
>
>
> > **Question 6.** Small corrections/suggestions
>
> Thank you so much for pointing out these typos. We will correct them in the revised version of the paper.

---

> > ### Comment · Reviewer_XEKF · 2025-11-21
> > **Response to authors**
> >
> > Thank you to the authors for the response and new SSCV evaluation. I am largely satisfied with the responses, with the exception of Question 3.
> >
> > First, I disagree that APS "aims to improve set size efficiency" --- that is what LAC does. It would be more correct to say that APS aims to improve set size _adaptivity_, which is closely related to conditional coverage. Here is a quote directly from the APS abstract "In this paper, we develop specialized versions of these techniques for categorical and unordered response labels that, in addition to providing marginal coverage, are also fully adaptive to complex data distributions, in the sense that they perform favorably in terms of approximate conditional coverage compared to alternative methods."
> >
> > Second, do you do the same sort of hyperparameter tuning for RAPS as you do for SOCOP? Is this a fair comparison?
> >
> > If my concerns regarding Question 3 can be addressed, and if the revisions mentioned in the author's response are adequately incorporated in an updated draft for us to see, I would be happy to raise my score.

---

> ### Author Response · Authors · 2025-11-21
>
> Thank you so much for your feedback. We completely agree that APS is not explicitly optimized for set-size efficiency, whereas the follow-up work, RAPS, introduces regularization to achieve this. We also agree that both APS and RAPS aim to improve set size adaptivity as a proxy of conditional coverage as well. Therefore, we deleted the sentence "We find
> this result remarkable..." and added a new subsection to show the SSCV evaluation results.
>
> Regarding hyperparameter tuning, we strictly follow the same tuning protocol for RAPS as in their original paper, using the grid $\lambda \in\\{0.001,0.01,0.1,0.2,0.5\\}$ to optimize set size, and a grid with smaller values $\lambda \in\\{0.00001,0.0001,0.0008,0.001,0.0015,0.002\\}$ to optimize SSCV. Similarly, for our SOCOP, we also use smaller $\lambda$ values (from a linearly spaced grid of 15 values over [0.05, 1.0] for optimizing the balance between set size and non-singleton rate to [0.005, 0.1] for optimizing SSCV), in order to achieve comparable SSCV.
> It is worth noting that in general smaller $\lambda$ will achieve smaller SSCV but worsen set size for both RAPS and SOCOP.
>
> However, from the results, we can see that
> our SOCOP method achieves a comparable SSCV while maintaining a reasonably
> small average size and a significantly lower non-singleton probability, demonstrating that
> SOCOP can achieve adaptivity without sacrificing efficiency.
>
> We have updated the paper with all the changes we have mentioned in all of our responses. For reference, we have added
> the intuition for  computing the nonconformity score on line 252; we deleted the sentence "We find
> this result remarkable..." on line 365; we added
> clarification for ImageNet-V2 on line 368; we add SSCV results in Sec. 3.1.2.
>
> Please let us know if there is any other concern. We are happy to address them.

---

### Official Review · Reviewer_5adu · 2025-11-01

**Soundness:** 3
**Presentation:** 2
**Contribution:** 3
**Rating:** 6
**Confidence:** 3

**Summary:**

The paper proposes a new nonconformity score with the goal of reducing the probability of producing non-singleton sets. The score is motivated by an optimization problem that optimizes a combination of the singleton objective and the expected length of the prediction sets subject to coverage constraint. The authors provide an efficient algorithm through a geometric reformulation to compute the nonconformity score. The paper includes empirical evaluation on image classification and LLM multiple-choice question answering that shows the proposed method reduces average set size and increases the frequency of singleton predictions.

**Strengths:**

The motivation behind the problem of singleton sets is important in practical settings. The paper presents a theoretically sound and efficient algorithm for the problem. Empirical evaluation shows that the method reduces the non-singleton probability while maintaining the average set size. Additionally, the extensions discussed in Section 2.2 are appreciated.

**Weaknesses:**

1. The paper is not written very clearly and it is hard to follow. The notation is extensive and dense in Section 2, with several new symbols introduced with little intuition. Additionally, the definition of nested conformal prediction was deferred for later, while the concepts are used in the paper; e.g., l215 ‘the nested sets property implies that…’ – this hampers the readability of the paper. It is suggested to work on this in the revised version.
2. While the paper guarantees marginal validity, there is no discussion on the implications of this objective on conditional coverage. Such analysis (theoretical and empirical) is warranted to understand the underlying tradeoffs.
3. Apart from Coverage and Avg Size, the empirical evaluation should report metrics that measure adaptivity of the sets e.g., worst-slice coverage, SSCV. The current evaluation is limited in its scope of studying the Avg Size and P(Size > 1) that are directly optimized by the method. To understand the usefulness of the sets, the analysis will benefit from additional metrics.

**Questions:**

Comments/typos:

- P4 l201: extra‘that’
- P5 l231: extra ‘the’

---

> ### Author Response · Authors · 2025-11-19
>
> ## Weaknesses
>
> > **Weakness 1.** The paper is not written very clearly and it is hard to follow. The notation is extensive and dense in Section 2, with several new symbols introduced with little intuition. Additionally, the definition of nested conformal prediction was deferred for later, while the concepts are used in the paper; e.g., l215 ‘the nested sets property implies that…’ – this hampers the readability of the paper. It is suggested to work on this in the revised version.
>
> Thank you so much for your suggestion to improve the paper. We are more than happy to take all measures to work on this in the revision. Regarding the specific comment that you raised, "nested sets" in line 215 does not actually refer to nested conformal prediction (which is indeed introduced later), despite the similarity in the names. Instead, it refers to the more common sense notions that the sets are included in each other. This has been mentioned and introduced in line 168, and its proof has been provided in the appendix. However, we understand that since it has not been stated as a lemma or definition in the text, it is easy to miss. We apologize for the misunderstanding. Therefore, we will move back the lemma to the main text.
>
> Moreover, in the revised version, given that we have an additional space page, we will add more textual explanation interspersed with the mathematical definitions. If you have any other specific suggestions, we are happy to incorporate them.
>
>
> > **Weakness 2.** While the paper guarantees marginal validity, there is no discussion on the implications of this objective on conditional coverage. Such analysis (theoretical and empirical) is warranted to understand the underlying tradeoffs.
>
> Thank you for this insightful suggestion. You are correct that our current submission focuses on split conformal prediction that targets to achieve marginal coverage. However, our proposed SOCOP non-conformity score is designed to be modular. While our paper implements it using standard split conformal prediction, the score itself can be seamlessly integrated into existing frameworks that aim for conditional coverage.
>
> For example, our SOCOP score $r(x,y)$ could be directly used as the input non-conformity score $S(x,y)$ in recent work [1]. Specifically, our SOCOP score $r(x,y)$ can be plugged into their augmented quantile regression objective (their Equation (2.4)). The resulting prediction set (defined by their Equation (2.5)) would then aim to satisfy approximate conditional coverage (Theorem 2), while simultaneously inheriting SOCOP's optimization goal of prioritizing singleton sets.
>
> Further, we have added below results with other metrics targeting conditional coverage.
>
> [1] Isaac Gibbs, John J Cherian, and Emmanuel J Cand`es. Conformal prediction with conditional guarantees. Journal of the Royal Statistical Society Series B: Statistical Methodology, page qkaf008, 2025.

---

> ### Author Response · Authors · 2025-11-19
>
> > **Weakness 3.** Apart from Coverage and Avg Size, the empirical evaluation should report metrics that measure adaptivity of the sets e.g., worst-slice coverage, SSCV. The current evaluation is limited in its scope of studying the Avg Size and $P(Size> 1)$ that are directly optimized by the method. To understand the usefulness of the sets, the analysis will benefit from additional metrics.
>
> Thanks for this valuable comment. We currently report the coverage, average size and $P(\text{size}>1)$ as they are the target metrics that our score aims to optimize. We are happy to discuss the metrics proposed by the reviewer for assessing the adaptivity of the prediction sets.
>
>  1. Worst-slice coverage (WSC) is introduced in [2, 3] as a proxy for evaluating conditional coverage. The WSC metric involves an adversarial search for the "worst" data slice $S_{v,a,b} = \\{x : a \le v^T X \le b\\}$. In these early works, the feature vectors $X$ in their experiments usually have a small dimension, such as a few hundreds for MNIST. For example, [3] first applies PCA to reduce the image's dimension to $p=50$, and approximates the search by randomly sampling 1000 independent direction vectors $v \in \mathbb{R}^p$ and searching over $a$ and $b$ along these directions. But the datasets used in our work, such as ImageNet, have a very high-dimensional feature space (over $150,000$), which makes both PCA and searching for the worst slice computationally intractable. Therefore, we instead turn to the next possible metric.
>
>
>  2. Size-stratified coverage violation (SSCV) is introduced in [4] as a measure of adaptiveness and conditional coverage violation. Following [4], we use the same set-size strata and tune the hyperparameter $\lambda$ to minimize SSCV on tuning set. The results of Resnet152-v2 and ViT-h-14 on ImageNet-Val are:
>
> | Method         | Coverage     | Avg Size       | P(Size>1)    | SSCV         |
> | -------------- | ------------ | -------------- | ------------ | ------------ |
> | Plug-in        | 0.969 ±0.003 | 47.362 ±7.138  | 0.469 ±0.025 | 0.046 ±0.001 |
> | RAPS           | 0.950 ±0.002 | 8.568 ±1.580   | 0.448 ±0.012 | 0.031 ±0.011 |
> | Pure Singleton | 0.950 ±0.002 | 250.539 ±4.554 | 0.250 ±0.005 | 0.050 ±0.000 |
> | LAS            | 0.950 ±0.002 | 2.279 ±0.046   | 0.467 ±0.007 | 0.197 ±0.026 |
> | SOCOP (ours)   | 0.950 ±0.002 | 3.372 ±0.198   | 0.304 ±0.008 | 0.039 ±0.009 |
>
> | Method         | Coverage     | Avg Size       | P(Size>1)    | SSCV         |
> | -------------- | ------------ | -------------- | ------------ | ------------ |
> | Plug-in        | 0.976 ±0.001 | 8.529 ±0.805   | 0.356 ±0.008 | 0.048 ±0.002 |
> | RAPS           | 0.950 ±0.002 | 7.652 ±2.259   | 0.319 ±0.007 | 0.047 ±0.003 |
> | Pure Singleton | 0.950 ±0.003 | 136.219 ±4.980 | 0.135 ±0.005 | 0.050 ±0.000 |
> | LAS            | 0.950 ±0.002 | 1.291 ±0.011   | 0.224 ±0.006 | 0.126 ±0.119 |
> | SOCOP (ours)   | 0.950 ±0.002 | 1.519 ±0.068   | 0.155 ±0.006 | 0.041 ±0.016 |
>
> Results are averaged over 100 random data splits. Please note that for the SSCV evaluation, we followed the protocol in [4] by selecting $\lambda$ from a grid of small values (specifically, smaller than the candidates used to optimize average size) to minimize SSCV on the tuning set. It is worth noting that while [4] reports these optimized SSCV figures, they do not report the corresponding average set sizes, which are typically much larger under this specific optimization setting (increases from $\approx3.2$ to $\approx8.6$ for Resnet152-v2 and from $\approx1.4$ to $\approx7.7$ for ViT-h-14).
>
> In contrast, **our SOCOP method achieves a comparable SSCV, while maintaining a reasonably small average size and a significantly lower non-singleton probability**, demonstrating that SOCOP can achieve adaptivity without sacrificing efficiency.
>
> [2] Maxime Cauchois, Suyash Gupta, and John C Duchi. Knowing what you know: valid and validated
> confidence sets in multiclass and multilabel prediction. Journal of machine learning research, 22(81): 1–42, 2021.
>
> [3] Yaniv Romano, Matteo Sesia, and Emmanuel Candes. Classification with valid and adaptive coverage.
> Advances in neural information processing systems, 33:3581–3591, 2020.
>
> [4] Anastasios Nikolas Angelopoulos, Stephen Bates, Michael Jordan, and Jitendra Malik. Uncertainty
> sets for image classifiers using conformal prediction. In International Conference on Learning
> Representations, 2021. URL https://openreview.net/forum?id=eNdiU_DbM9.
>
>
> ## Questions
> > **Question 1.** Comments/typos:
>
> Thank you so much for pointing out these typos. We will correct them in the revised version of the paper.

---

> > ### Comment · Reviewer_5adu · 2025-11-28
> >
> > I thank the authors for their detailed response.
> >
> > A couple of comments and suggestions:
> >
> > - Thanks for sharing the SSCV results. While I do not expect the authors to report new results with WSC, I would like to point that PCA with the logits layer for ImageNet is feasible and prior works have adopted a similar method.
> >
> > - I would appreciate if the final version of this paper includes more discussion on the conditional coverage implications. Achieving conditional coverage inherently requires the set size to be large on certain regions of the feature space, which is at odds with the singleton optimization problem.
> >
> > Overall, I remain positive of the work as I was earlier and believe that it meets the acceptance threshold.

---

> > > ### Author Response · Authors · 2025-11-29
> > >
> > > Thanks so much for your feedback!
> > >
> > > 1. We agree that one can approximate WSC on ImageNet by applying PCA to hidden representations or logits. However, this introduces a model-induced bias. WSC is defined by searching over slabs in the original feature space of $X$; when we work with hidden representations or logits, we are defining slabs in model-dependent proxy feature spaces, and different architectures induce different proxy feature spaces even on the same dataset. This is another reason why we choose to evaluate and report SSCV, where we demonstrate that SOCOP can achieve adaptivity without sacrificing efficiency.
> > > 2. We have updated our Sec. 3.4 Discussion to include more discussion on conditional coverage in our revised version.
> > >
> > > Thanks again for your support of this work! We are grateful for your valuable expertise.
> > >
> > > Sincerely, The authors

---

### Author Response · Authors · 2025-12-01
**Summary of Rebuttal**

We thank the reviewers for their constructive feedback. We believe we have addressed all the concerns, weaknesses, questions and comments. Below, we summarize how our rebuttal addresses the main points raised by reviewers.

1. **Conditional Validity and Adaptiveness.** A common concern (Reviewer 5adu, XEKF, b3RD) was the lack of metrics evaluating conditional coverage and adaptivity. We added additional experiments for the Size-Stratified Coverage Violation (SSCV) metric on ImageNet, which demonstrates that SOCOP can achieve adaptivity without sacrificing efficiency. In addition, we clarified that our SOCOP score is modular. We discussed how it can be seamlessly integrated into existing conditional conformal frameworks to provide theoretical conditional coverage guarantees.

2. **Comparisons with Additional Baselines and Frameworks.** Reviewer b3RD asked for a comparison with existing efficiency-oriented frameworks. We clarified the fundamental difference between our method and the existing efficiency-oriented frameworks (in short, we introduce a new score for an objective different than that considered in prior works). Furthermore, we added additional experiments to compare our method against the aggregated score baseline suggested by the reviewer, showing that our SOCOP significantly outperforms these baselines.

3. **Justification of the Singleton Objective.** Reviewers (Reviewer  b3RD, 67Nw) asked for a stronger justification for prioritizing singletons in prediction. We clarified that our objective derives from the M-criterion (Vovk et al, 2005), and expanded the discussion on the operational value of singleton predictions and the critical distinction from a standard Top-1 prediction.

4. **Analysis of Effect on P(size=0).** Reviewers b3RD and 67Nw inquired about the effect on P(size = 0). We clarified the theoretical conditions under which empty sets occur: while impossible when $\lambda=0$, they effectively serve as valid abstentions when $\lambda>0$ and model confidence is low. Empirically, we demonstrated that empty sets occur with negligible frequency ($< 0.01\%$) across all experiments.

5. **Clarity and Writing.** We clarified that our paper already provides a detailed analysis of role and effect of the  hyperparameter $\lambda$, and highlighted our strict data-split protocol to address concerns regarding tuning bias (Reviewer 67Nw). Furthermore, we restructured the paper to improve readability and rigor. Key revisions include moving the "nested sets" lemma to the main text (addressing Reviewer 5adu's comment), adding intuitions for computing the nonconformity score, clarifying distribution shifts in ImageNet-V2 (addressing Reviewer XEKF's comment), and restructuring Section 2 with two explicit subsections (addressing Reviewer 67Nw's comment)

6. **Future Direction.** We discussed how our SOCOP can be integrated into class-conditional frameworks (Mondrian/clustered CP). Additionally, we outlined future theoretical pathways for designing scores intrinsically targeted for conditional coverage and exploring data-efficient hyperparameter selection strategies while accounting for tuning bias.

Finally, although the scores have been reverted to the initial state, we want to highlight that during the discussion period, **Reviewer b3RD raised the score to 6 (Nov 25)**, and **Reviewer 67Nw raised the score to 8 (Nov 26)**. Additionally, **on Nov 21, Reviewer XEKF stated they would be "happy to raise" the score** pending the resolution of Question 3. We have updated the manuscript to fully address this question by: 1) correcting the distinction between set-size efficiency and adaptivity regarding APS/RAPS (removing the unwarranted claim), 2) adding the requested SSCV evaluation, and 3) clarifying that we followed a strictly comparable hyperparameter tuning protocol for RAPS and our SOCOP to ensure a fair comparison.
Thus, **we believe that a reasonable expectation for the final scores would be 6/8/6/8**.

---

### Meta-Review · Area_Chair_Dfcb · 2026-01-07

**Summary:**

1. Clarity and Readability: Multiple reviewers noted dense notation, large text blocks, deferred definitions (e.g., nested conformal prediction), lack of intuition/diagrams for score computation, and minor typos. These made sections harder to follow.
2. Justification and Motivation for Singleton Objective: Questions about whether prioritizing singletons is universally beneficial (application-dependent), need for more concrete practical examples/citations, and clearer distinction from top-1 predictions or existing efficiency frameworks.
4. Conditional Coverage and Adaptivity: The method guarantees only marginal coverage, not conditional (unlike baselines like APS/RAPS). Limited initial discussion of implications and lack of adaptivity metrics (e.g., worst-slice coverage or Size-Stratified Coverage Variance - SSCV).
4. Empirical Evaluation Limitations: Insufficient metrics beyond average size and P(size>1); sensitivity to hyperparameter λ tuning (potential bias); unclear comparisons to non-conformal baselines (e.g., Plug-In) or why certain datasets (e.g., ImageNet-V2) were challenging; imprecise handling of edge cases like empty sets (P(size=0)).
5. Comparisons and Extensions: Uncertainty about advantages over adaptive/efficient methods; limited elaboration on class-conditional extensions or integration with conditional frameworks.

Authors effectively addressed these via additions like SSCV experiments (showing good adaptivity), conditional coverage discussions, practical justifications with citations, λ sensitivity analyses, writing restructurings (subsections, moved lemmas), and edge-case analyses (empty sets negligible). This resolved most issues, with three reviewers raising scores (one to 8, others to 6), supporting a positive outlook.

**Reviewer Concerns:**

Addressed Concerns: See Summary

Outstanding concerns:

Worst-Slice Coverage (WSC) Evaluation (Raised by Reviewer 5adu): Suggested using PCA for feasibility on ImageNet, but authors declined due to computational and bias concerns, leaving this unexplored.

**Reviewer Scores:**

All reviewers tended to provide a positive rating.

---

### Decision · Program_Chairs · 2026-01-26

Accept (Poster)